# EMFuse: Energy-based Model Fusion for Decision Making

**Kejie He**[1],[*] **Yi-Chen Li**[1],[*]
**Yang Yu**[1],[†]
[1]National Key Laboratory for Novel Software Technology, Nanjing University, Nanjing, China
& School of Artificial Intelligence, Nanjing University, Nanjing, China
{hekj,liyc}@lamda.nju.edu.cn
yuy@lamda.nju.edu.cn

## Abstract

Model fusion has emerged as a promising research direction, offering a resource-efficient paradigm that leverages existing pre-trained models to circumvent the need for training from scratch. In this work, we investigate the fusion of models specifically adapted for decision-making tasks. This challenge divides into two distinct, yet related subproblems: the direct fusion of models that act as policy and the fusion of dynamics models that subsequently induce a policy. We suggest that these seemingly divergent subproblems can be unified through the lens of energy-based models (EBMs), which parameterize a conditional distribution via an energy function where lower energy implies higher probability. Our framework, **EMFuse**, provides this convergence by leveraging the concept of energy as a common currency for fusion. For direct fusion of policies, such as those in language models, the output distribution is commonly softmax (Boltzmann), which essentially defines the negative logarithmic probability as an energy function. For dynamics models, existing works often train a set of models on the same dataset to obtain robust uncertainty estimation; such an ensemble approach leads to an exponential explosion in computational complexity when it comes to dynamics fusion across multiple sets of models. To overcome this, we introduce the Any-step Dynamics Energy-based Transition Model (ADETM), a novel architecture that performs efficient single-model-per-dataset uncertainty estimation with its energy-based backbone, thereby avoiding this computational explosion. Our EMFuse framework surpasses other baselines by 0.34% to 6.63% on single/cross domain discrete decision-making benchmarks, and achieved an extra 2.3 to 7.4 normalized points on average in D4RL MuJoCo continuous-control scenarios. Our code is available at https://github.com/LAMDA-RL/EMFuse.

## 1 Introduction

Modern deep learning ecosystems are increasingly populated by highly specialized models that power tasks from text generation to complex problem-solving (Alto, 2024; Wu et al., 2024). Directly training new monolithic models is expensive, while discarding existing expertise is wasteful. *Model fusion* promises a resource-efficient alternative by combining pre-trained experts into a stronger system (Wortsman et al., 2022; Matena & Raffel, 2022; Hu et al., 2021; Lu et al., 2024). However, while model fusion has seen rapid growth, its application to the specialized domain of decision-making remains comparatively underexplored. Recognizing the immense potential in this area, our work focuses specifically on *fusion for decision making*, a critical frontier for creating more capable and adaptable intelligent agents from fusion (Levine et al., 2020; Moerland et al., 2023).

At a high level, the behavior of a decision making agent is governed by either a directly learned policy or a policy derived from a learned dynamics model. This fundamental architectural choice presents two distinct points of intervention for model fusion: (1) *Direct policy fusion:* This approach

---

[*]Equal Contribution
[†]Corresponding Author

involves combining the output distributions of multiple policies at the point of decision, potentially deriving a single, more robust action by taking all distributions into consideration. (2) *Dynamics fusion:* In model-based RL (MBRL), dynamics models learned from offline logs (Levine et al., 2020) enable further policy training (Chua et al., 2018; Janner et al., 2019; Yu et al., 2020; Hafner et al., 2020). The dynamics fusion approach acts at a more foundational level by merging the agent's dynamics model (their predictive understanding of the environment's dynamics). The goal is to construct a single, more comprehensive and reliable simulation of the transitions by unifying each model's knowledge of the world. A superior policy could be learned under this unification, which provides a richer and more accurate basis for decision-making.

These seemingly divergent subproblems can be unified. We take the perspective of Energy-Based Models (EBMs), where distributions are represented through energies whose additive composition corresponds to multiplicative densities (LeCun et al., 2006; Hinton, 2002). In this view, both policy outputs and dynamics likelihoods can be written as $p(y \mid x) = \exp(-E(x, y))/Z(x)$, and fusion becomes an *energy sum* $E_{\text{fuse}} = \sum_i \lambda_i E_i$. This makes policy fusion and dynamics fusion two instances of the same principle, differing only by the choice of $x, y$ and the sampler.

Our framework, **EMFuse**, applies energy-additive fusion to two key settings: (1) direct policy fusion and (2) dynamics model fusion. This approach enables efficient and effective inference without requiring additional training of the full models. We observe that the fused energy distribution remains close to each constituent policy on its specific domain, as measured by small KL-divergence. This faithfulness makes the fused energy a powerful indicator for selecting the best policy for a given context, a principle we formalize in our **EMSelect** algorithm.

Most existing dynamics model learning work trains an ensemble of models for uncertainty estimation (Yu et al., 2020). However, fusing ensembles of dynamics models quickly becomes intractable due to exponential blow-up. Inspired by ADMPO (Lin et al., 2025) that enables uncertainty estimation on one model by its ability to predict multiple next state with flexible action sequence input, we therefore introduce the *Any-step Dynamics Energy-based Transition Model (**ADETM**)*, an energy-based dynamics model that performs single-model-per-dataset uncertainty estimation while retaining multi-step context. Additional state and action encoders are added to make ADETM available to the energy-based context. This setup avoids the combinatorial cost of cross-ensemble fusion.

**Our main contributions are listed as follows:**

- **EMFuse:** We formalize policy and dynamics fusion under the same energy framework $E_{\text{fuse}} = \sum_i \lambda_i E_i$, connecting classical PoE (Hinton, 2002) to decision-making settings.

- **EMSelect:** An EMFuse-based policy selection framework that achieved additional **1.18%-1.31%** gain on top of EMFuse.

- **ADETM for tractable dynamics fusion:** A single-model-per-dataset energy-based world model with any-step context that circumvents cross-ensemble explosion while preserving uncertainty-aware behavior.

- **Empirical gains:** On single/cross-domain discrete decision-making benchmarks, EMFuse improves accuracy by **0.34%–6.63%**, and on D4RL MuJoCo continuous control (Fu et al., 2020) it adds **+2.3 to +7.4** normalized points on average over other fusion baselines.

## 2 PRELIMINARIES

### 2.1 TOWARDS ENERGY-BASED MODELS FOR DECISION MAKING

**Markov Decision Process** We consider a discounted MDP $\mathcal{M} = (\mathcal{S}, \mathcal{A}, P, r, \gamma)$ with an offline dataset $\mathcal{D} = \{(s, a, r, s')\}$ collected by an unknown behavior policy $\pi_\beta$ (Puterman, 1994; Sutton & Barto, 2018; Levine et al., 2020). Offline training faces a *support gap*: test-time states/actions may lie outside the empirical support of $\mathcal{D}$, causing distribution shift and value overestimation. This makes calibrated *uncertainty* and *support awareness* central to algorithm design.

**Behavior modeling: explicit vs. implicit** One axis models or constrains by the behavior distribution. The *Explicit* approaches fit an estimate $\hat{\pi}_\beta(a \mid s)$ and use them as a prior or regularizer for

policy learning (Fujimoto et al., 2019; Kostrikov et al., 2022). *Implicit* approaches shape learning objectives to bias solutions toward the dataset support, e.g., conservative value learning and pessimistic objectives (Kumar et al., 2020; Levine et al., 2020). Both aim to remain out-of-distribution (OOD)-robust; they differ in how support is represented (density vs. objective penalties).

**Dynamics Models and Uncertainty Estimation.** A second axis learns a model of the dynamics and plans or trains policies inside it. Classical likelihood models parameterize $p_\theta(s' \mid s, a)$ with Gaussian or mixture families; latent/variational world models compress trajectories into learned state abstractions (e.g., *World Models*, PlaNet, Dreamer) (Ha & Schmidhuber, 2018; Hafner et al., 2019; 2020). Recent work also explores diffusion/score-based generative modeling for sequential control (Janner et al., 2022). Likelihood models require calibration under off-distribution conditions; consequently, robust uncertainty estimation becomes key.

Epistemic uncertainty is commonly estimated through bootstrapped or independently initialized model ensembles; aleatoric uncertainty is captured by predictive dispersion (Kendall & Gal, 2017; Chua et al., 2018). ADMPO (Lin et al., 2025) trains a single recurrent net on variable-length transitions, achieving ensemble-like robustness without model repetition, further simplifying the framework. Ensembles not only improve robustness, they also provide *composable* uncertainty signals that will align naturally with an energy-based view.

**Energy Based Models (EBMs) .** EBMs represent distributions via unnormalized energies, $p_\theta(x) \propto \exp\big( - E_\theta(x) \big)$, trained with contrastive or score-based criteria (Hinton, 2002; Hyvärinen, 2005; LeCun et al., 2006). Energies add *linearly* when combining independent experts, corresponding to multiplicative composition of their unnormalized densities. Specializing to dynamics, Energy-based Transition Models (**ETMs**) (Chen et al., 2024) learn a transition energy $E_\theta(s, a, s')$ whose negative exponent defines a next-state distribution,

$$p_\theta(s' \mid s, a) \ \propto \ \exp\big( - E_\theta(s, a, s') \big), \tag{1}$$

enabling support-aware modeling and contrastive learning of transitions. Behavior priors also admit an energy view, $E_\beta(s, a) = - \log \pi_\beta(a \mid s)$. Given expert energies $\{E_i\}$, *Product-of-Experts* fusion corresponds to an *energy sum*

$$E_{\text{fuse}}(x) \ = \ \sum_i \lambda_i E_i(x), \qquad p_{\text{fuse}}(x) \ \propto \ \exp\big( - E_{\text{fuse}}(x) \big), \tag{2}$$

which can combine multiple transition experts (ensembles, domains) and behavior priors in a single, support-conscious objective (Hinton, 2002; LeCun et al., 2006). This additive structure underpins the tractable fusion rules we use later for both policy and dynamics.

**Policy learning** ETMs or fused ETMs can be used to generate rollouts for policy learning under pessimism/regularization (Rubinstein & Kroese, 2004; Chua et al., 2018; Feinberg et al., 2018; Buckman et al., 2018; Kumar et al., 2020). This closes the loop in offline MBRL: behavior priors constrain actions, world models predict consequences with quantified uncertainty, and additive energies provide a principled path to compose experts while remaining support-aware.

## 2.2 BOLTZMANN OUTPUT AS ENERGIES

Autoregressive policies, such as those in modern LLMs, can be viewed directly through an energy-based lens. The key prerequisite for fusing such policies is that they must operate over a shared, finite vocabulary $\mathcal{V}$, a condition typically met by using a common tokenizer (Vaswani et al., 2017; Brown et al., 2020; Hoffmann et al., 2022; Touvron et al., 2023). This shared support makes their output distributions directly comparable and fusible.

At each step $t$, the model maps a context $x_{\leq t}$ to a vector of logits $z_t \in \mathbb{R}^{|\mathcal{V}|}$. These logits define a normalized next-token policy via the softmax function with temperature $\tau$:

$$p_\theta(y_t \mid x_{\leq t}) = \text{softmax}\big(\tfrac{1}{\tau} z_t\big) = \frac{\exp\big(z_t(y)/\tau\big)}{\sum_{y' \in \mathcal{V}} \exp\big(z_t(y')/\tau\big)}. \tag{3}$$

Trained via maximum likelihood, this is the canonical estimator for conditional token probabilities (Bishop, 2006; Goodfellow et al., 2016). Critically, it is equivalent to a Boltzmann(softmax) distribution with energy $E_\theta(x_{\leq t}, y) = -z_t(y)/\tau$. This direct equivalence—where log-probabilities

are scaled negative energies—is the cornerstone that allows us to apply our energy-additive fusion framework to LLM policies.

# 3 ENERGY-BASED MODEL FUSION FOR DECISION MAKING

In this section, we will mainly discuss applications of our **E**nergy-based **M**odel **Fus**ion for **D**ecision Making (**EMFuse**) framework to direct policy fusion and dynamics fusion. A subsequent application of EMFuse in policy selection (**EMSelect**) will be mentioned and the supporting architecture that enables dynamics training (**ADETM**) will be introduced.

The property of energy additivity forms the core of our fusion framework (Hinton, 2002), which we now define formally. Let $\{E_i(x,y)\}_{i=1}^n$ be the energy of the policy model / dynamic model that defines normalized conditionals $p_i(y \mid x) = \exp\big(-E_i(x,y)\big)/Z_i(x)$ on a shared support. For nonnegative weights $\lambda_i$ with $\sum_i \lambda_i = 1$, define the fused energy

$$E_{\text{fuse}}(x,y) \;=\; \sum_{i=1}^n \lambda_i\, E_i(x,y). \tag{4}$$

Then the fused distribution is

$$p_{\text{fuse}}(y \mid x) \;=\; \frac{\exp\big(-E_{\text{fuse}}(x,y)\big)}{Z_{\text{fuse}}(x)} \;\propto\; \prod_{i=1}^n p_i(y \mid x)^{\lambda_i}, \tag{5}$$

i.e., a *logarithmic opinion pool* (geometric mixture, or **LogOP**), which is the unique minimizer of the weighted reverse-KL projection $\arg\min_q \sum_i \lambda_i \mathrm{KL}(q\|p_i)$ (Heskes, 1998; Genest & Zidek, 1986). This rule is application-agnostic; subsequent subsections instantiate $E_i$ for (i) energy-based policies (e.g. LLMs; via Eq. 3) and (ii) dynamics models (e.g. ETMs; Eq. 1).

## 3.1 DIRECT POLICY FUSION

By taking an energy-based perspective, many recent decision-making models can be written as the conditional distribution: $p_\theta(y \mid x) = \exp(-E_\theta(x,y))/Z_\theta(x)$. Whenever experts expose such normalized policies on a shared support, **EMFuse** fuses them by *adding energies* (equivalently, multiplying their distributions): $E_{\text{fuse}} = \sum_i \lambda_i E_i \iff p_{\text{fuse}}(y \mid x) \propto \prod_i p_i(y \mid x)^{\lambda_i}$, which is precisely the logarithmic opinion pool (LogOP)—the optimizer of a weighted reverse-KL projection and coincides with a Product-of-Experts (PoE) in probability space (Heskes, 1998; Genest & Zidek, 1986; Hinton, 2002; LeCun et al., 2006).

We extend this to a real case-study by taking modern LLMs as an energy-based policy example. Contemporary LLMs are decoder-only Transformers trained with next-token prediction (Vaswani et al., 2017; Brown et al., 2020; Hoffmann et al., 2022; Touvron et al., 2023). By equation 3, at time $t$, each expert $M_i$ induces energies $E_i(x_{\leq t}, y) = -z_i(y)/\tau_i$ over the shared vocabulary $\mathcal{V}$, so $\log p_i(\cdot \mid x_{\leq t})$ are (negative) energies up to a normalizer.

Maximum-likelihood training makes Eq. 3 the canonical estimator of conditional token probabilities (Bishop, 2006; Goodfellow et al., 2016). Equivalently, this is a Boltzmann distribution with energy $E_\theta(x_{\leq t}, y) = -z_t(y)/\tau$, so $\log p_\theta$ is (negative) energy up to a normalization constant—exactly the form required for energy-additive fusion.

Let experts $\{M_i\}_{i=1}^n$ share (or be mapped to) the same tokenizer, hence the same support $\mathcal{V}$. Each produces $p_i(y \mid x_{\leq t}) = \text{softmax}(z_i/\tau_i)$. EMFuse defines the fused policy $p_{\text{fuse}}$ as the LogOP solution

$$p_{\text{fuse}}(\cdot \mid x_{\leq t}) = \arg\min_{q \in \Delta(\mathcal{V})} \sum_{i=1}^n \lambda_i\, \mathrm{KL}(q \,\|\, p_i(\cdot \mid x_{\leq t})) \implies p_{\text{fuse}}(y \mid x_{\leq t}) \propto \prod_{i=1}^n p_i(y \mid x_{\leq t})^{\lambda_i}, \tag{6}$$

with nonnegative weights $\lambda_i$ summing to 1. Writing $p_i = \exp(-E_i)$ yields *energy additivity* $E_{\text{fuse}} = \sum_i \lambda_i E_i$, i.e., multiplicative densities with additive energies (Hinton, 2002; LeCun et al., 2006). Check Algorithm 1 for details, experiments w.r.t. LLM settings will be discussed in §4.

---

**Algorithm 1** EMFuse-Policy (Case study on LLM)

---

1: **Input:** context $x_{\leq t}$; experts $M_{1:n}$; per-expert temperatures $\{\tau_i\}$;
   weights $\{\lambda_i\}$ ($\lambda_i \geq 0$, $\sum_i \lambda_i = 1$); optional decoding (temperature/top-$k$/top-$p$).
2: **For** $i = 1..n$**:** compute logits $z_i \leftarrow M_i.\text{logits}(x_{\leq t})$; log-probs $\ell_i \leftarrow \text{logsoftmax}(z_i/\tau_i)$
3: **Fuse in log-space:** $\ell_{\text{fuse}} \leftarrow \sum_{i=1}^{n} \lambda_i \ell_i$
4: **Normalize:** $p_{\text{fuse}} \leftarrow \exp\big(\ell_{\text{fuse}} - \text{logsumexp}(\ell_{\text{fuse}})\big)$
5: **Decode:** sample/argmax $y_t \sim p_{\text{fuse}}(\cdot \mid x_{\leq t})$; append $y_t$ to context.
6: **Output:** fused next-token distribution (and the generated token if decoding).

---

The following discussions should be noticed: fusion operates over an identical support $V$; this avoids vocabulary mismatch and makes Eq. 6 well-defined. In practice, this condition can be satisfied either by sharing a tokenizer or by mapping heterogeneous vocabularies into a common space. Recent work on vocabulary mapping and tokenizer transplantation (Sennrich et al., 2016; Kudo & Richardson, 2018; Mavromatis et al., 2024; Xu et al., 2024; Goddard & Neto, 2025) shows that such projections can be implemented in a training-free or lightly fine-tuned way. While a detailed engineering treatment is beyond our scope, EMFuse is designed to operate downstream of these mapping layers: as long as the mapping preserves semantic monotonicity (i.e., avoids severe semantic drift), the PoE structure in Eq. 6 remains valid and empirically robust. Crucially, the additive energy structure ($E_{\text{fuse}} = \sum_i E_i$) provides inherent resilience to mapping artifacts. If a vocabulary projection introduces uncertainty (scale distortion), it naturally manifests as a higher-entropy (flatter) energy surface. In a Product-of-Experts, these flattened distributions are automatically down-weighted in favor of the sharper, more confident energies from native experts, effectively filtering out mapping noise without manual intervention. For weights, $\lambda_i$ can be static (uniform/held-out tuned) or context-adaptive (e.g., entropy-aware); mild per-expert temperatures $\tau_i$ adjust sharpness without changing the EMFuse form (Guo et al., 2017; Hinton et al., 2015).

In practice, we are applying uniform weights, and ablation studies on entropy-based weight showed statistically insignificant results, which will be discussed in the Appendix §E. Apply decoding heuristics (temperature, top-$k$, top-$p$) post-process $p_{\text{fuse}}$ but do not alter its normalized form (Holtzman et al., 2020). We observed minimal, statistically insignificant gains from adding Laplace smoothing of per-expert distributions (to prevent multiplicative collapse when a policy assigns zero probability to a viable token); details and ablations in Appendix §E.1. From the LogOP perspective, EMFuse acts as a conservative consensus or "AND-gate" over experts: tokens that any expert assigns very low probability are exponentially suppressed in the fused distribution. As a result, a single miscalibrated expert cannot destabilize the decision rule. This stability guard is verified in the distribution faithfulness experiment in Table 2 and further increases our EMSelect design in the following discussion.

## 3.2 Direct Policy Fusion - An alternative

EMFuse provides a consensus distribution by additive energies (Eq. 4–5). Empirically, when experts are domain-specialized, this consensus tends to lie close (in KL) to the expert whose domain dominates the current context. This suggests using EMFuse *as a reference* to decide which expert should act at each decoding step, which is the heart of our **EMSelect** algorithm.

**Two-expert derivation.** Consider two experts $i$ and $j$ with token policies $p_i(\cdot \mid x_{\leq t})$ and $p_j(\cdot \mid x_{\leq t})$. Instantiate EMFuse *on this pair* with weights $\alpha \in [0, 1]$ and $1 - \alpha$ (defaults to $\alpha = \frac{1}{2}$):

$$p_{i \oplus j}(\cdot \mid x_{\leq t}) \; \propto \; p_i(\cdot \mid x_{\leq t})^{\alpha} \, p_j(\cdot \mid x_{\leq t})^{1-\alpha}.$$

Select the expert whose policy is closer (smaller KL) to this pairwise fused reference:

$$\text{choose } i \; \text{iff} \; \text{KL}\big(p_{i \oplus j} \, \| \, p_i\big) \; \leq \; \text{KL}\big(p_{i \oplus j} \, \| \, p_j\big).$$

Since $\text{KL}(p \| q) = \mathbb{E}_p[\log p - \log q]$, the entropy term cancels and the decision reduces to

$$\mathbb{E}_{p_{i \oplus j}}[\log p_i] \; \geq \; \mathbb{E}_{p_{i \oplus j}}[\log p_j],$$

i.e., pick the expert with higher expected log-likelihood under the pairwise EMFuse reference.

**EMSelect (tournament over $n$ experts).** For $n \geq 2$, apply the two-expert selector sequentially in a lightweight tournament *without* any initial seeding: (i) fix a deterministic order over experts (e.g., their indices), (ii) set the *incumbent* to the first expert in that order, (iii) compare the incumbent against the next expert using the two-expert rule, (iv) keep the winner as the new incumbent, and (v) continue until all experts have been compared once.

---

**Algorithm 2** EMSelect: KL-guided per-step policy choice from EMFuse

---

1: **Input:** context $x_{\leq t}$; experts $M_{1:n}$; pairwise weight $\alpha \in [0, 1]$ (default $\alpha = \frac{1}{2}$).
2: **Expert log-probs:** for all $i$, compute $\ell_i \leftarrow \log p_i(\cdot \mid x_{\leq t})$ (via logits$\rightarrow$log-softmax).
3: **Initialize incumbent:** let the comparison order be $(1, 2, \ldots, n)$; set $i^\star \leftarrow 1$.
4: **for** $j = 2$ **to** $n$ **do**
5:    **Pairwise EMFuse:** $\ell_{i^\star \oplus j} \leftarrow \alpha \, \ell_{i^\star} + (1 - \alpha) \, \ell_j$;   $p_{i^\star \oplus j} \leftarrow \mathrm{softmax}(\ell_{i^\star \oplus j})$
6:    **Score experts:** $S(i^\star) \leftarrow \sum_y p_{i^\star \oplus j}(y) \, \ell_{i^\star}(y)$;   $S(j) \leftarrow \sum_y p_{i^\star \oplus j}(y) \, \ell_j(y)$
7:    **Advance winner:** $i^\star \leftarrow \arg\max_{k \in \{i^\star, j\}} S(k)$
8: **end for**
9: **Decode:** use $p_{i^\star}(\cdot \mid x_{\leq t})$ to sample/argmax $y_t$; append to context.
10: **Output:** selected expert $i^\star$ and per-step policy $p_{i^\star}$.

---

**EMSelect complements EMFuse.** EMFuse aggregates all experts into a conservative consensus, while EMSelect commits at each decoding step to the expert that best preserves that consensus *locally*. In domains with strong specialization and mild calibration differences, this targeted commitment yields the additional accuracy reported in our benchmarks. **Theoretical guarantee.** Let $y_{1:T}$ denote a generated token sequence of length $T$. For each expert $i$, let $p_i(\cdot \mid x_{\leq t})$ be its conditional distribution given context $x_{\leq t}$ (which includes prompt $x_0$ and history $y_{<t}$), and let $p_{\mathrm{fuse}}(\cdot \mid x_{\leq t})$ be the EMFuse consensus. EMSelect induces a per-step distribution $p_{\mathrm{sel}}(\cdot \mid x_{\leq t}) = p_{i_t}(\cdot \mid x_{\leq t})$ by selecting an index $i_t$ at each step, and we denote the resulting sequence distributions by $p_{\mathrm{sel}}(y_{1:T})$ and $p_{\mathrm{fuse}}(y_{1:T})$. For conciseness, define the worst-case local expert-to-fuse divergence

$$\Delta_t^{\max}(x_{\leq t}) := \max_i \mathrm{KL}\big(p_i(\cdot \mid x_{\leq t}) \,\|\, p_{\mathrm{fuse}}(\cdot \mid x_{\leq t})\big).$$

Appendix E.4 shows, via the KL chain rule for autoregressive policies, that

$$\mathrm{KL}\big(p_{\mathrm{sel}} \,\|\, p_{\mathrm{fuse}}\big) \; \leq \; \sum_{t=1}^{T} \mathbb{E}_{y_{<t} \sim p_{\mathrm{sel}}}\big[\Delta_t^{\max}(x_{\leq t})\big]. \tag{7}$$

Empirically, all expert distributions remain very close to the fusion mean (per-step divergences satisfy KL $< 0.09$, see Table 2), so the upper bound in Eq. 7 is tight in our setting. Thus, EMSelect is mathematically "leashed" to the conservative EMFuse geometry: it can exploit sharper experts locally, yet any sequence-level deviation is constrained by the shared consensus.

### 3.3 DYNAMICS MODEL FUSION

Recall the general EMFuse rule (Eqs. 4–5): expert energies add and their (unnormalized) densities multiply. Specializing to Energy-based Transition Models (ETMs; Eq. 1), a set of $n$ expert transitions $\{E_i(s, a, s')\}_{i=1}^{n}$ defines the fused transition

$$E_{\mathrm{fuse}}(s, a, s') \; = \; \sum_{i=1}^{n} \lambda_i \, E_i(s, a, s'), \qquad p_{\mathrm{fuse}}(s' \mid s, a) \; \propto \; \exp\big(-E_{\mathrm{fuse}}(s, a, s')\big), \tag{8}$$

with nonnegative weights $\lambda_i$ (we use $\lambda_i = \frac{1}{n}$ by default), assuming shared state/action spaces and possibly different training datasets/domains per expert. In continuous state spaces, normalization is unnecessary for ranking or sampling procedures that operate in energy/log-space.

**ADETM: any-step ETM enabling single-model uncertainty.** We employ an *Any-step Dynamics Energy-based Transition Model* (**ADETM**) to retain ETM's training recipe while equipping each expert with robust, ensemble-like uncertainty at *single-model* cost. This is inspired by the ADMPO (Lin et al., 2025) framework. Concretely, ADETM wraps the ETM energy network with (i) an MLP *state encoder* and (ii) a multi-head-attention *action-sequence encoder* operating on a fixed

---

**Algorithm 3** EMFuse–Dynamics (one step)

---

1: **Input:** current state $s_t$; action $a_t$; expert ETM energies $\{E_i\}_{i=1}^n$; fusion weights $\lambda_i = \frac{1}{n}$.
2: **Per-expert energies:** compute $E_i(s_t, a_t, \cdot)$ for all $i$ (on candidate $s'$ or as an energy field).
3: **Fuse in energy space:** $E_{\text{fuse}}(s_t, a_t, \cdot) \leftarrow \sum_{i=1}^n \lambda_i E_i(s_t, a_t, \cdot)$.
4: **Sample/score next state:** draw $s_{t+1} \sim p_{\text{fuse}}(\cdot \mid s_t, a_t)$ via Langevin dynamics on $E_{\text{fuse}}$ (as in ETM), or take a MAP estimate $s_{t+1} = \arg\min_{s'} E_{\text{fuse}}(s_t, a_t, s')$ if sampling is not used.
5: **Output:** $s_{t+1}$ (optionally, diagnostics derived from $E_{\text{fuse}}$).

---

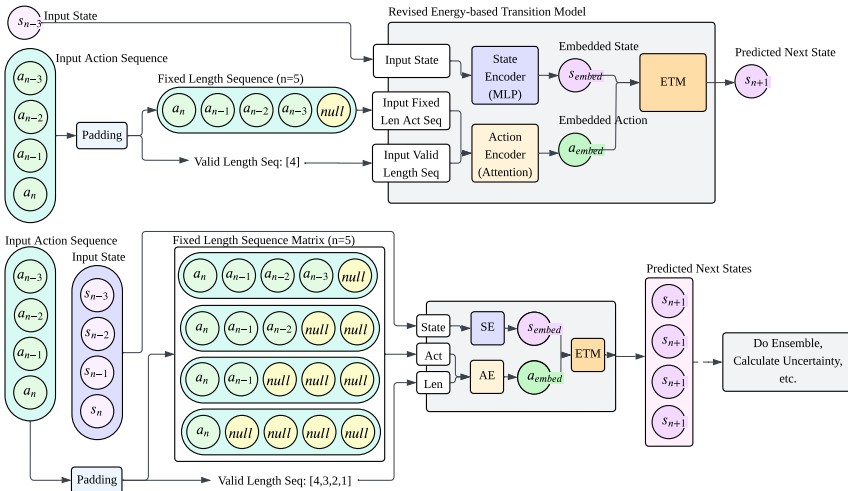

Figure 1: **A**ny-step **D**ynamics **E**nergy-based **T**ransition **M**odel for dynamics fusion 3.3. Top: single-step next-state prediction; bottom: stacked-input variant for parallel multi-step prediction.

history window $k$ with a valid-length mask. The encoders produce a joint embedding $\begin{bmatrix} h_s \parallel h_a \end{bmatrix}$ that conditions the ETM energy $E_\theta(s, a_{t-k:t}, s')$. This preserves the ETM module and its training dynamics (contrastive / InfoNCE objective, Langevin sampling for training and diagnostics), while allowing ADETM to exploit short-horizon action context without duplicating models. Figure 1 illustrates the architecture; full encoder details and hyperparameters are deferred to the appendix.

**Uncertainty without ensembles via stacked histories.** ADETM yields a practical history-sensitivity regularizer by *stacking* feasible history slices from a FIFO queue of the last $k$ state–action pairs and measuring dispersion across the resulting next-state predictions. Let the queue contain $(s_{t-4}, a_{t-4}), \ldots, (s_t, a_t)$. For the same target step $t+1$, construct up to $k$ valid history slices

$$(s_{t-4}, a_{t-4:t}) \rightarrow \hat{s}_{t+1}^{(1)}, \ (s_{t-3}, a_{t-3:t}) \rightarrow \hat{s}_{t+1}^{(2)}, \ \ldots, \ (s_t, a_t) \rightarrow \hat{s}_{t+1}^{(k)}.$$

Define the uncertainty score as the sample variance (or mean-squared dispersion) across predictions,

$$u_\theta(s_t, a_t) \;=\; \frac{1}{k} \sum_{m=1}^k \big\| \hat{s}_{t+1}^{(m)} - \overline{s}_{t+1} \big\|_2^2, \qquad \overline{s}_{t+1} \;=\; \frac{1}{k} \sum_{m=1}^k \hat{s}_{t+1}^{(m)}. \tag{9}$$

This dispersion behaves analogously to ensemble disagreement but requires only one trained ADETM per expert, effectively solving the exponential explosion when fusing ensembled models. Consequently, EMFuse rollouts with ADETM scale with the number of experts rather than the ensemble size, yielding a lightweight dynamics module in terms of parameters, FLOPs, and roll-out latency; we provide a quantitative analysis in Appendix B.5 and ablate the resulting uncertainty signal against risk-neutral and local-sensitivity baselines in Appendix B.6. The fused transition $p_{\text{fuse}}(s' \mid s, a)$ and ADETM's uncertainty are then utilized within an offline RL loop to generate model rollouts and train a policy; we defer training protocols and all evaluations to the Experiments section and the appendix.

## 4 EXPERIMENTS

### 4.1 EVALUATION OVERVIEW

We evaluate EMFuse on two complementary LLM families and on model-based offline control. **Family L (Llama, Touvron et al. (2023))** is used to assess *mechanistic fidelity* via length-controlled preference (§C.3) and token-level KL to domain experts (§C.4). **Family Q (Qwen Yang et al. (2024); OpenCompass Contributors (2023))** is used to assess *task accuracy* on finance/mathematics/medication under the standard OpenCompass protocol (§C.2). This design highlights model-agnostic benefits (accuracy under an external evaluator; Family Q) while establishing distributional faithfulness under controlled tokenization/training (Family L).

### 4.2 RESEARCH QUESTIONS

**RQ1 (Policy fusion effectiveness).** Does EMFuse improve decision performance over parameter-space and regularized merging baselines? *Metrics:* OpenCompass accuracy (Family Q; Table 20); AlpacaEval length-controlled win rate (Family L; Tables 16, 17).

**RQ2 (Faithful distribution).** Does EMFuse preserve each expert's distribution on its own domain? *Metric:* token-level $D_{\mathrm{KL}}(1\mathrm{B}_{\mathrm{expert}} \parallel \cdot)$ under teacher forcing (Family L; Table 18).

**RQ3 (Selection vs. fusion).** Does EMSelect (KL-guided tournament) provide additional gains over EMFuse? *Metrics:* same as RQ1 (deltas over EMFuse; see §3.2 and aggregated gains reported).

**RQ4 (Dynamics fusion).** Does EMFuse with ADETM improve offline RL returns versus dynamics-level baselines? *Metric:* D4RL normalized return (IQM) with BCa 95% CIs (Table 19).

### 4.3 MODELS, DATA, AND TASKS

**Family L (Llama).** We train $1\mathrm{B}$ SFT experts on *language* (harmless/helpful) and *subject* (agriculture/medication/philosophy) splits; $8\mathrm{B}$ counterparts follow a FLOPs-matched budget (Eq. 10). Full training and generation settings are in §B.2. **Family Q (Qwen).** We SFT mixed-subject experts (finance/mathematics/medication) and specific-subject (within finance, 3 different categories) §B.1 ; Evaluation is done with OpenCompass defaults (§C.2). **Dynamics.** ADETM (§3.3) is trained on D4RL-v2; rollout and SAC details are in §B.4.

### 4.4 BASELINES

We compare to three representative training-free baselines spanning the main fusion axes:

- **Uniform Model Soup** (Wortsman et al., 2022): a *data-less, parameter-space* merger widely adopted in practice; it requires no access to the original training data and provides a reference point for weight-space averaging.
- **RegMean** (Nguyen et al., 2025): a *training-free* (and data-less) regularized weight merging method that aligns statistics without task data; this family is commonly identified as a canonical training-free approach in recent overviews.
- **PackLLM** (Mavromatis et al., 2024): a *training-free, logit-space* policy fusion method at inference time. PackLLM's pairwise packing informed our EMSelect tournament design (§3.2). This baseline is exclusive to the policy fusion.

Together these baselines test EMFuse against: (i) parameter-space averaging (Soup), (ii) trainless weight alignment (RegMean), and (iii) output/logit-space fusion (PackLLM).

### 4.5 EVALUATION PROTOCOLS

**Family L (mechanistic fidelity).** We use AlpacaEval with length control to mitigate verbosity bias (§C.3) and measure token-level $D_{\mathrm{KL}}$ from each $1\mathrm{B}_{\mathrm{expert}}$ to EMFuse and its $8\mathrm{B}$ counterpart under a shared tokenizer (§C.4), isolating distributional faithfulness from capacity. **Family Q (task accuracy).** We report OpenCompass accuracy on finance/mathematics/medication using framework

defaults (§C.2). **Dynamics.** We report D4RL normalized returns as IQM with BCa confidence intervals under identical SAC/ADETM settings.

**Reporting.** RL results use IQM with BCa 95% CIs over $N=5$ seeds. LLM results follow the same protocol: AlpacaEval uses paired bootstrap over prompts; OpenCompass uses its default resampling.

## 4.6 RESULTS SUMMARY AND VALIDITY CONSIDERATIONS

**RQ1/RQ3 — Policy fusion effectiveness.** On OpenCompass, **EMFuse** improves over parameter-space and logit-space training-free baselines on both aggregates reported in Table 1. On the *subject-mix*, EMFuse attains $63.49^{+1.23}_{-1.23}$, outperforming Soup (60.88) and RegMean (60.31), and narrowly exceeding PackLLM (63.15). On the *finance-suite*, EMFuse reaches $89.21^{+0.01}_{-1.40}$ versus 88.27 (Pack-LLM), 83.51 (Soup), and 82.58 (RegMean). Task-level details are in Appendix Tables 20 and 21. Complementary preference evaluations on **Family L** (AlpacaEval with length control) show EM-Fuse strongly outperforming Soup/RegMean and remaining competitive with larger 8B variants on some subject splits; see Appendix Tables 16–17 for per-dataset win rates and CIs. Appendix B.6 further disentangles the role of ADETM's temporal-consistency uncertainty signal by comparing it against a risk-neutral variant and a local-sensitivity (action-noise) variant under the same rollout and SAC configuration. Using EMFuse as a reference for per-step selection further improves accuracy. From Table 1, **EMSelect** gains $+1.31$ points on the subject-mix and $+1.18$ on the finance-suite over EMFuse. Per-task breakdowns in Appendix Tables 20 and 21 show improvements concentrated in finance (e.g., FPB $+2.07$, LendingClub $+1.42$) and medication (e.g., MedQAM $+2.68$), with small regressions on some math sets (e.g., MGSMZ $-2.40$), yielding positive aggregate deltas.

Table 1: **OpenCompass (Family Q) — average accuracy.** Two aggregates are shown: *subject-mix* (finance/mathematics/medication) and *finance-suite*. Numbers are averages with 95% CIs ($^{+u}_{-l}$). See Appendix Tables 20 and 21 for per-task breakdowns as tables are too large to fit in this section.

| Aggregate | Soup | RegMean | PackLLM | EMFuse (Ours) | EMSelect (Ours) |
|---|---|---|---|---|---|
| Subject-mix | $60.88^{+1.25}_{-1.25}$ | $60.31^{+1.25}_{-1.25}$ | $63.15^{+1.24}_{-1.24}$ | $\mathbf{63.49^{+1.23}_{-1.23}}$ | $\mathbf{64.80^{+1.25}_{-1.25}}$ |
| Finance-suite | $83.51^{+0.94}_{-0.79}$ | $82.58^{+0.91}_{-0.85}$ | $88.27^{+1.42}_{-0.04}$ | $\mathbf{89.21^{+1.40}_{-0.01}}$ | $\mathbf{90.39^{+1.39}_{-0.07}}$ |

Table 2: **Distributional faithfulness (Family L).** Average token-level Kullback–Leibler divergence from each 1B domain expert ($1B_{exp}$) to **EMFuse** (**EM-F**) and 8B experts on five evaluation domains (columns), **smaller is better**, with 95% confidence intervals. Domain descriptions: "harmless" and "helpful" refer to RLHF-style safety and alignment sets; "agriculture", "medication", and "philosophy" refer to subject-specific QA and reasoning benchmarks. Evaluation details in Appendix §C.4.

| Comparison Pairs | Language-based domains | | Subject-specific domains | | |
|---|---|---|---|---|---|
| | **harmless** | **helpful** | **agriculture** | **medication** | **philosophy** |
| $\mathbf{1B_{exp} \rightarrow EM\text{-}F}$ | $\mathbf{0.0391^{+0.0015}_{-0.0014}}$ | $\mathbf{0.0801^{+0.0035}_{-0.0034}}$ | $\mathbf{0.0485^{+0.0027}_{-0.0025}}$ | $\mathbf{0.0481^{+0.0014}_{-0.0014}}$ | $\mathbf{0.0459^{+0.0005}_{-0.0005}}$ |
| $1B_{exp} \rightarrow 8B$ | $0.5063^{+0.0191}_{-0.0190}$ | $0.3075^{+0.0124}_{-0.0121}$ | $0.5880^{+0.0190}_{-0.0177}$ | $0.2827^{+0.0079}_{-0.0081}$ | $0.2041^{+0.0022}_{-0.0022}$ |

**RQ2 — Faithful distribution on experts' home domains.** Table 2 shows that the token-level $D_{KL}(1B_{expert} \parallel \cdot)$ from each 1B domain expert to **EMFuse** is consistently small ($\approx 0.04$–$0.08$), and markedly below the divergence to its 8B counterpart ($\approx 0.20$–$0.59$). This supports that EMFuse preserves domain-specific token probabilities more faithfully than capacity scaling.

**RQ4 — Dynamics fusion for offline control.** Table 3 reports D4RL IQM returns: **EMFuse** averages **50.1** across Hopper/Walker2d/HalfCheetah, exceeding our data-free (Soup) and training-free (RegMean) baselines, and slightly surpassing the Mixed-data oracle baseline (47.8) on average. These findings support the conclusion that EMFuse, together with ADETM, provides a tractable policy training framework that not only sidesteps the exponential overhead of naïve ensembling but also preserves or enhances the performance of domain-matched experts across control tasks.

**Scope and caveats.** (1) *Evaluator heterogeneity.* Family Q uses accuracy-based OpenCompass, while Family L uses preference-based AlpacaEval (mitigated via length control); we therefore avoid

Table 3: Offline-RL performance on the MuJoCo medium quality benchmark (5 seeds, evaluation over the last 100 steps). Higher IQM return is better. Bootstrap CI provided on the right-hand side.

| Environment | EMFuse (Ours) | RegMean | Soup | Mixed (Oracle) |
|---|---|---|---|---|
| Hopper | $\mathbf{49.03}^{+0.66}_{-0.47}$ | $46.34^{+3.44}_{-2.68}$ | $47.33^{+3.19}_{-3.01}$ | $49.35^{+2.73}_{-1.06}$ |
| Walker2d | $\mathbf{59.53}^{+7.13}_{-6.87}$ | $52.24^{+12.29}_{-8.18}$ | $46.52^{+12.44}_{-19.31}$ | $51.64^{+16.24}_{-28.64}$ |
| HalfCheetah | $\mathbf{41.83}^{+1.06}_{-1.98}$ | $32.80^{+0.33}_{-2.89}$ | $34.36^{+2.38}_{-3.92}$ | $42.48^{+0.53}_{-1.37}$ |
| AVERAGE | $\mathbf{50.1}$ | $43.8$ | $42.7$ | $47.8$ |

cross-family numeric comparisons and interpret each within its protocol. (2) *Tokenizer/control of confounds.* KL analyses are restricted to Family L to ensure shared tokenization; extending KL to other families is left for future work due to compute limits. (3) *Uncertainty in RL.* Offline-RL results show wide CIs in some environments; we emphasize aggregate IQM and provide per-env CIs (Table 3) to avoid over-interpreting single tasks. (4) *Selection trade-offs.* EMSelect's gains are not uniform across tasks (notably some math sets); we attribute this to local selection being most beneficial when expert specialization is strong and calibration is comparable, consistent with the per-task deltas in Appendix Tables 20 and 21. (5) *Scope of baselines.* We focus on widely used training-free mergers (Soup/RegMean) and a logit-space method (PackLLM) to match our operating regime; parameter-tuned or data-hungry mergers are out of scope. (6) *Weight-related ablations.* We use uniform fusion weights by default; entropy-based reweighting yielded statistically insignificant differences, and Laplace smoothing produced negligible gains. Check Appendix §E.1.

## 5 CONCLUSION

In this work, we introduced EMFuse, a principled framework that unifies model fusion for decision-making under the lens of energy-based models. By treating energy as a common currency, our approach recasts the seemingly disparate problems of direct policy fusion and dynamics model fusion as two instances of a single, elegant principle: the additive composition of energies.

Our core contributions are synergistic. The EMFuse framework provides the theoretical foundation, while the ADETM architecture makes this theory practical for modern offline reinforcement learning by circumventing the intractable computational cost of fusing model ensembles. Furthermore, our investigation reveals a fundamental trade-off between two distinct fusion philosophies: consensus versus commitment. EMFuse, as a logarithmic opinion pool, generates a conservative consensus distribution, hedging against the failure of any single expert. In contrast, EMSelect implements a higher-variance commitment strategy, making a winner-take-all decision at each step. Our empirical results underscore this trade-off; while EMSelect excels when a context aligns perfectly with one expert's specialty, the consensus approach of EMFuse proves more robust on complex reasoning tasks where a synthesis of diverse perspectives is superior to a single, potentially flawed, viewpoint.

While our current study was scoped to a shared-vocabulary setting due to computational constraints, we view the challenge of aligning heterogeneous tokenizers as a tractable engineering problem rather than a fundamental barrier, especially given recent successes in vocabulary mapping (Mavromatis et al., 2024; Xu et al., 2024; Goddard & Neto, 2025). Integrating these techniques will unlock the full potential of applying EMFuse to diverse, off-the-shelf models. Overall, EMFuse provides a solid and extensible foundation for collaborative AI, demonstrating that the simple addition of energies is a powerful and versatile tool for decision-making.

ACKNOWLEDGMENTS

This work was supported by the Yangtze River Delta Science and Technology Innovation Community Joint Research Program 2024CSJZN00302, the National Science Foundation of China (62495093,62495090), and the Natural Science Foundation of Jiangsu (BK20243039). The authors thank anonymous reviewers for their helpful discussions and suggestions for improving the article.

ETHICS STATEMENT

This paper proposes an efficient model fusion strategy and conforms with the ICLR Code of Ethics in every respect.

REPRODUCIBILITY STATEMENT

This paper provides all the information needed to reproduce the main results in the appendix.

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

## A    LLM USAGE STATEMENT

In compliance with the ICLR 2026 Code of Ethics, we disclose the use of Large Language Models (LLMs) in this work. Our usage is detailed as follows:

1. **LLMs as Research Subjects:** The core of our experimental evaluation involved using various large language models (e.g., Llama and Qwen families) to generate outputs for benchmarking purposes. All performance metrics and subsequent analyses presented in Section 4 are derived from the outputs of these models.

2. **LLMs as an Assisting Tool:** A large language model was used for minor stylistic improvements and to assist in generating the LaTeX code for some of the tables presented in the manuscript.

The authors have meticulously reviewed and verified all content, including any text or code assisted by an LLM, to ensure its correctness and factual accuracy. The authors take full responsibility for the entire content of this submission.

## B    POLICY TRAINING DETAILS

### B.1    TRAINING THE QWEN FAMILY OF EXPERTS (THE Q FAMILY)

We instantiate **ALL Q** experts from **Qwen 2.5 7B**(Yang et al., 2024; Team, 2024) using supervised fine-tuning (SFT) with LoRA (Hu et al., 2021). Following a uniform recipe, we apply instruction-style formatting (answer-only loss), cosine LR with warm-up, and LoRA on all linear submodules. For each adapter we train multiple runs (varying seed and a small LoRA grid) and *select the best performing checkpoint* by validation loss and downstream OpenCompass score (§C.2). To standardize training length, we use the largest epoch budget *per group*: **4** epochs for *Subject-Mix* and **10** epochs for the *Finance-suite*. This setup aims to explore cross domain (Subject-Mix) and within domain (Finance-suite) fusion results, with each post-SFT policy relatively strong in their own expertise, as we can see in Table 20 and 21. All training regarding the Qwen family was conducted on a single A100 node containing 4 GPUs.

**Adapters and training sets.**    Table 4 lists the exact adapter names (as shown in the results tables) and their paired training sets. Evaluation benchmarks are summarized separately in Table 5.

Table 4: Q-family adapters and their training sets (Qwen 2.5 7B base). Best checkpoint per adapter is selected from multiple runs.

| Group | Adapter name | Training set(s) |
|---|---|---|
| Subject-Mix | **Fin**ance | CRA LendingClub(Ariza-Garzón et al., 2024) |
| | **Math**ematics | Orca Math Word Problems (Mitra et al., 2024) |
| | **Med**ication | MedQA + MedMCQA (Jin et al., 2020; Pal et al., 2022) |
| Finance-suite | **FPB** | Financial PhraseBank (Malo et al., 2014) |
| | **Head**lines | Financial news headlines (Sinha & Khandait, 2020) |
| | **Lend**ingClub | LendingClub (TheFinAI, 2024; Ariza-Garzón et al., 2024) |

**Evaluation benchmarks (OpenCompass).**    We evaluate with OpenCompass defaults under fixed decoding and normalization rules (cf. §C.2), macro-averaging within domains. Benchmarks per domain are:

**Decoding and score selection.**    OpenCompass decoding parameters are left at framework defaults for comparability. For each adapter we keep the checkpoint with the lowest validation loss; ties are broken by the higher domain score under the corresponding OpenCompass evaluator.

Table 5: Evaluation benchmarks used for the **Q** family.

| Domain | Benchmark (full name) | abbrv. |
|---|---|---|
| Finance (Subject-Mix) | Financial Opinion Mining & QA—Sentiment Analysis | FiQASA |
| | LendingClub approval / credit risk | LendingClub |
| Mathematics (Subject-Mix) | Grade School Math (8K) | GSM8K |
| | Multilingual GSM (English) | MGSME |
| | Multilingual GSM (Chinese) | MGSMZ |
| Medication (Subject-Mix) | MedQA (Mainland/CN/ZH) | MedQAM |
| | MedQA (Taiwan/CN/ZH) | MedQAT |
| | MedQA (USMLE/US/EN) | MedQAU |
| | MedMCQA (Medical MCQ benchmark) | medmcqa |
| Finance-suite | Financial PhraseBank (AllAgree subset) | FPB |
| | Financial news headline sentiment | Headline |
| | LendingClub approval / credit risk | LendingClub |

Table 6: Canonical SFT hyper-parameters for **Subject-Mix** adapters (Qwen 2.5 7B).

| Parameter | Value |
|---|---|
| Base model | `Qwen2.5-7B` |
| Epochs (`num_train_epochs`) | **4** |
| Per-device train batch size | 2 |
| Gradient accumulation steps | 8 |
| Learning rate | $2 \times 10^{-5}$ |
| Weight decay | 0 |
| Max grad norm | 1 |
| Warmup ratio | 0.03 |
| LR scheduler | `cosine` |
| LoRA rank (`lora_r`) | 64 |
| LoRA alpha (`lora_alpha`) | 128 |
| LoRA dropout | 0 |
| LoRA target modules | all linear layers |
| Context length (`max_length`) | 8192 |
| Loss masking | answer tokens only |

Table 7: Canonical SFT hyper-parameters for **Finance-suite** adapters (Qwen 2.5 7B).

| Parameter | Value |
|---|---|
| Base model | `Qwen2.5-7B` |
| Epochs (`num_train_epochs`) | **10** |
| Per-device train batch size | 2 |
| Gradient accumulation steps | 4 |
| Learning rate | $2 \times 10^{-5}$ |
| Weight decay | 0 |
| Max grad norm | 1 |
| Warmup ratio | 0.03 |
| LR scheduler | `cosine` |
| LoRA rank (`lora_r`) | 32 |
| LoRA alpha (`lora_alpha`) | 64 |
| LoRA dropout | 0 |
| LoRA target modules | all linear layers |
| Context length (`max_length`) | 4096 |
| Loss masking | answer tokens only |

## B.2 TRAINING THE LLAMA FAMILY OF EXPERTS (THE L FAMILY)

We instantiate all **L** experts from the **Llama 3.x** line using supervised fine-tuning (SFT) on shared tokenizers so that all policies operate over an identical vocabulary $\mathcal{V}$ (cf. §2.2). Unless otherwise noted, we perform *full-parameter* SFT (no LoRA), adopt answer-only loss masking, and use standard `transformers+DeepSpeed` training (§B.2.2). For robustness, we train multiple runs (varying seed and modest optimizer/schedule settings) and *select the best checkpoint* by validation loss and downstream metrics (length-controlled AlpacaEval win rate; KL to domain experts; see §C.3, §C.4). To match compute across capacities, the 8B expert training is *FLOPs-capped* to the aggregate of the 1B experts as in Eq. 10 (within 1%).

$$\sum_{i=1}^{n} \text{FLOPs}(1\text{B}_i) \approx \text{FLOPs}(8\text{B}), \qquad (10)$$

**Experts and training sets.** We organize **L** experts into two groups—*Language Quality* (helpfulness/harmlessness) and *Subject Knowledge* (agriculture/medication/philosophy). Table 8 enumerates each expert, its base model, and the corresponding training set(s).

Table 8: L-family experts and their training sets. The 8B experts are trained under a FLOPs budget matched to the aggregate 1B experts (Eq. 10).

| Group | Expert | Base model | Training set(s) |
|---|---|---|---|
| Language | $1\text{B}_{\text{base}}$ | Llama 3.2-1B | — |
| | $1\text{B}_{\text{harmless}}$ | Llama 3.2-1B | RLHF-harmless (Bai et al., 2022; Ouyang et al., 2022) |
| | $1\text{B}_{\text{helpful}}$ | Llama 3.2-1B | RLHF-helpful (Bai et al., 2022; Ouyang et al., 2022) |
| Subject | $1\text{B}_{\text{base}}$ | Llama 3.2-1B | — |
| | $1\text{B}_{\text{agriculture}}$ | Llama 3.2-1B | Agriculture-QA (KisanVaani, 2024) |
| | $1\text{B}_{\text{medication}}$ | Llama 3.2-1B | Consumer Health QA (Savery et al., 2020) |
| | $1\text{B}_{\text{philosophy}}$ | Llama 3.2-1B | Ethical Problem-Solving (Corrêa et al., 2024) |
| Aggregate | $8\text{B}_{\text{language}}$ | Llama 3.1-8B | RLHF-harmless (Bai et al., 2022; Ouyang et al., 2022) RLHF-helpful (Bai et al., 2022; Ouyang et al., 2022) |
| | $8\text{B}_{\text{subject}}$ | Llama 3.1-8B | Agriculture-QA (KisanVaani, 2024) Consumer Health QA (Savery et al., 2020) Ethical Problem-Solving (Corrêa et al., 2024) |

**Canonical SFT hyper-parameters.** To keep this section parallel to the Qwen presentation, we summarize the SFT settings used for 1B and FLOPs-capped 8B Llama experts in two compact tables (Meta, 2024a;b). These consolidate the Trainer/DeepSpeed configuration into the key choices that most affect optimization and throughput. (Full details remain in §B.2.2.)

Table 9: Canonical SFT hyper-parameters for **Llama 1B** experts.

| Parameter | Value |
|---|---|
| Base model | `Llama-3.2-1B` |
| Epochs (`num_train_epochs`) | 3 |
| Per-device train batch size | 4 |
| Gradient accumulation steps | 8 |
| Max sequence length | 256 |
| Optimizer | AdamW |
| LR scheduler | linear (WarmupDecayLR equiv.) |
| Precision | fp16 |
| Loss masking | answer tokens only |
| DeepSpeed | ZeRO-3 (off by default for 1B) |

**Evaluation protocol and checkpoint selection.** For **L**, we evaluate with AlpacaEval under length control (§C.3) and compute token-level $D_{\text{KL}}$ to domain experts (§C.4). Best checkpoints per expert

Table 10: Canonical SFT hyper-parameters for **Llama 8B** experts (FLOPs-capped to Eq. 10).

| Parameter | Value |
|---|---|
| Base model | `Llama-3.1-8B` |
| Epochs (`num_train_epochs`) | 3 (stop early on FLOPs target) |
| Per-device train batch size | 2 |
| Gradient accumulation steps | 8 |
| Max sequence length | 256 |
| Optimizer | AdamW ($\beta_1$=0.9, $\beta_2$=0.999, $\epsilon$=1e−8) |
| LR scheduler | linear (WarmupDecayLR equiv.) |
| Precision | fp16 |
| DeepSpeed | ZeRO-3 with CPU offload (params & optimizer) |
| Loss masking | answer tokens only |

are chosen by (i) lowest validation loss and (ii) highest downstream score on these evaluators (ties broken by the latter). Decoding uses a uniform configuration across models (Table 11); EOS or `max_new_tokens` terminates generation.

**Licenses.** All Llama checkpoints are retrieved via Hugging Face (Davison et al., 2019). Please review the associated Meta licenses before use (Meta Platforms, Inc., 2024b;a).

### B.2.1 LLM EXPERTS AND TRAINING SETS

Table 8 specifies base models and datasets for all **L** experts.

### B.2.2 TRAINING SETUP AND FLOPS PARITY

We train with `transformers`' `Trainer` and DeepSpeed ZeRO-3. The 8B experts use a `FLOPsStopCallback` to enforce the budget $\sum_i \text{FLOPs}(1B_i) \approx \text{FLOPs}(8B)$ (Eq. 10). Hardware (typical): 1B experts on one 4090 node (2 GPUs); 8B experts on one 4090 node (4 GPUs).

### B.2.3 GENERATION CONFIGS

We implement Algorithm 1 by computing per-expert log-probabilities (`log_softmax`), summing in log-space with weights $\lambda_i$, normalizing, then decoding. Each expert runs on a dedicated GPU; the fusion step executes on a single device.

Table 11: LLM generation configuration.

| Config | Value |
|---|---|
| `max_new_tokens` | 200 |
| `temperature` | 1.0 |
| `top_p` | 0.95 |

We terminate on EOS or `max_new_tokens`, applying the same decoding heuristics to the fused distribution as to single models (cf. §2.2).

### B.3 ADETM ARCHITECTURE AND TRAINING

We adopt the Energy-based Transition Model (ETM) formalism (Chen et al., 2024) and use the *Any-step Dynamics Energy-based Transition Model* (**ADETM**; cf. §3.3) to encode short action histories at *single-model* cost. Figure 2 shows the single-step and stacked multi-step variants.

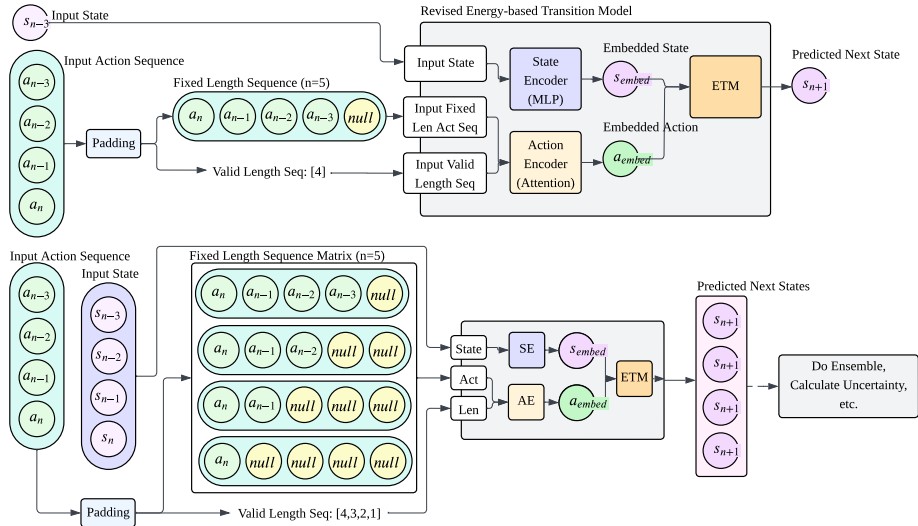

Figure 2: **ADETM** variants: top—single-step next-state prediction; bottom—stacked history for parallel multi-step predictions. Mirrors Fig. 1 in the main text.

**State encoder.** Given $s \in \mathbb{R}^{d_s}$,

$$h_s \; = \; \mathrm{LN}\big(\sigma(W_2\,\sigma(W_1 s + b_1)) + b_2\big) \in \mathbb{R}^{d_h},$$

with two fully connected layers $W_1 \in \mathbb{R}^{d_h \times d_s}$, $W_2 \in \mathbb{R}^{d_h \times d_h}$, ReLU activations, and LayerNorm.

**Action-sequence encoder.** Let $A \in \mathbb{R}^{B \times L_{\max} \times d_a}$ be padded action sequences with true lengths $\ell \in \{1, \ldots, L_{\max}\}^B$.

---

**Algorithm 4** ActionEncoder$(A, \ell)$

---

1: $X \leftarrow \mathrm{Linear}_{d_a \to d_h}(A)$          // projection
2: $X \leftarrow X + \mathrm{PosEmb}(0{:}L_{\max} - 1)$
3: $\mathrm{mask}_{ij} \leftarrow [\, j \geq \ell_i \,]$          // key-padding mask
4: $Y \leftarrow \mathrm{MHA}(X; \mathrm{mask})$
5: $Y \leftarrow \mathrm{LN}(Y + X)$
6: $Z \leftarrow \mathrm{FFN}(Y)$
7: $Z \leftarrow \mathrm{LN}(Z + Y)$
8: **return** $h_a = \frac{1}{\ell_i} \sum_{j < \ell_i} Z_{ij}$

---

The ADETM energy network conditions on $[h_s \,\|\, h_a]$.

**Environments and hyper-parameters.** We use D4RL-v2 (Fu et al., 2020) (Apache 2.0) across `Hopper`, `Walker2d`, and `HalfCheetah`, with *random/medium/expert* splits. Each split yields a fixed offline buffer; unless otherwise noted, training uses a single RTX 4090. ADETM defaults:

Table 12: ADETM training defaults. Entries above the midrule follow ETM (Chen et al., 2024); `embedding_hidden_dims` applies to both state and action encoders.

| Config | Default |
|---|---|
| `etm_lr` | $1 \times 10^{-3}$ |
| `etm_hidden_dims` | $[200, 200, 200, 200]$ |
| `etm_activation` | `relu` |
| `etm_with_reward` | `true` |
| `etm_softmax_temperature` | 1.0 |
| `etm_num_negative_samples` | 16 |
| `etm_loss_type` | `info_nce` |
| `etm_add_grad_penalty` | `true` |
| `etm_grad_penalty_margin` | 5.0 |
| `etm_langevin_iter` | 50 |
| `etm_max_epochs` | 500 |
| `etm_batch_size` | 1024 |
| `embedding_hidden_dims` | 256 |

### B.4 POLICY LEARNING (OFFLINE SAC IN WORLD MODELS)

We adopt Soft Actor–Critic (SAC) with two Q-networks (ensemble) (Haarnoja et al., 2018; An et al., 2021; Nikulin, 2023). Hyper-parameters are held fixed across environments; Table 13. Rollouts use ADETM with a short FIFO history (up to the maximum action-sequence length; cf. Fig. 2). At step $t$, we backtrack up to $\min(L_{\max}, t)$ and compute dispersion across stacked predictions to obtain the penalty signal (as in Eq. 9 in the main text).

Table 13: Offline SAC hyper-parameters for world-model experiments.

| Config | Default |
|---|---|
| `actor_lr` | $1 \times 10^{-4}$ |
| `critic_lr` | $3 \times 10^{-4}$ |
| `critic_nums` | 2 |
| `gamma` | 0.99 |
| `tau` | 0.005 |
| `alpha` | 0.2 |
| `alpha_lr` | $1 \times 10^{-4}$ |
| `rollout_freq` | 1,000 |
| `rollout_batch_size` | 5,000 |
| `rollout_length` | 15 |
| `penalty_coef` | $p$ |
| `model_retain_epochs` | 5 |
| `real_ratio` | 0.5 |
| `epoch` | 2,500 |
| `step_per_epoch` | 1,000 |
| `batch_size` | 256 |
| `eval_episodes` | 10 |
| `penalty_type` | `ensemble_std` |

We set $p=0.3$ for `Hopper` and $p=0.7$ for `HalfCheetah`/`Walker2d`. Seeds: $[1, 2, 3, 4, 5]$. A fixed budget of 2,500 steps is used; checkpoints are saved every 10 steps. Evaluation on the true `*-medium` environments uses 10 episodes per checkpoint. We report normalized return with inter-quartile mean (IQM) and BCa bootstrap 95% CIs across 5 seeds following Agarwal et al. (2022). ADETM hyper-parameters match Table 12 at rollout.

### B.5 Computational profile of ADETM

To make the computational footprint of ADETM explicit, we construct a controlled rollout-only simulation tailored to the default configuration used in our world-model experiments. Throughout the paper, ADETM uses a maximum valid history length of $k=5$, so that a single ADETM evaluation can condition on up to five recent state–action sequence pairs when forming its uncertainty signal. In the notation of the main text, ensemble-based world models with $M$ environments and $N$ dynamics models per ensemble require $O(M \times N)$ dynamics evaluations per uncertainty-aware rollout step, whereas ADETM replaces per-environment ensembles with a single transition network per environment, which scales as $O(M)$ (§3.3). To study this scaling under realistic architectures, we instantiate an ensemble configuration with $N=5$ dynamics models per environment, matching the default ADETM history length $k=5$.

Both ADETM and the $N=5$ ensemble reuse the same transition-network architecture as in our real runs, but gradient computation and parameter updates are disabled so that we measure only inference-time costs of the dynamics module. In a practical ensemble, these $N$ dynamics models would typically be initialized with different random seeds and/or trained on resampled data; here we deliberately keep architectures and weights identical across the $N$ ensemble members so that any differences arise purely from the number of dynamics evaluations per step rather than representational capacity. Each configuration is plugged into the same world-model-based SAC rollout code path as in §B.4 on the D4RL-v2 medium tasks, with identical history length and hyperparameters but with gradients turned off. The goal of this subsection is solely to characterize computational scaling of the dynamics module; all return metrics for our method are reported separately in the main Experiments section.

Table 14 reports (i) the number of trainable parameters in the dynamics module, (ii) the estimated floating-point operations (FLOPs) per epoch of this rollout-only procedure (using the same counting scheme as §B.4), and (iii) the measured wall-clock rollout latency in milliseconds per environment step on one 4090 node.

Table 14: Computational profile of ADETM versus an ensemble of $N=5$ dynamics models per environment on D4RL-v2 medium tasks in a rollout-only simulation. FLOPs are per epoch of the world-model-based SAC loop; latency is wall-clock milliseconds per rollout step.

| Environment | Dynamics setting | Params (M) | Epoch FLOPs (T) | Latency (ms/step) |
|---|---|---|---|---|
| Hopper | ADETM ($k=5$) | 0.69 | 12.32 | 569 |
| | Ensemble ($N=5$ models) | 3.47 | 44.35 | 642 |
| Walker2d | ADETM ($k=5$) | 0.70 | 12.43 | 589 |
| | Ensemble ($N=5$ models) | 3.48 | 44.62 | 669 |
| HalfCheetah | ADETM ($k=5$) | 0.70 | 12.43 | 591 |
| | Ensemble ($N=5$ models) | 3.48 | 44.62 | 673 |

Across all three tasks, ADETM reduces the number of dynamics parameters by a factor of $5\times$ (about 0.69–0.70M vs. 3.47–3.48M) and the epoch-level dynamics FLOPs by roughly $3.6\times$ (about 12.3–12.4T vs. 44.3–44.6T), while also providing a consistent reduction in rollout latency (approximately 10–14%). Because this simulation reuses the trained transition architecture and tensor shapes from our world-model runs and differs only in the number of dynamics evaluations per step, these results concretely instantiate the $O(M)$ vs. $O(M \times N)$ argument in §1 and §3.3.

## B.6 ABLATION ON ADETM UNCERTAINTY SIGNALS

To assess whether ADETM's performance gains stem from its temporal-consistency uncertainty signal rather than from a generic variance heuristic, we compare ADETM against two baselines that share the same dynamics model but modify the uncertainty term used in the offline RL penalty.

**Baseline A: Risk-neutral (Zero Penalty).** This baseline utilizes the full ADETM dynamics model for rollout generation but eliminates the pessimism term from the reward function. The penalized reward $\tilde{r}(s,a)$ is computed as

$$\tilde{r}(s,a) = \hat{r}(s,a) - 0 \cdot u_\theta(s,a), \tag{11}$$

i.e., the penalty coefficient is explicitly set to $\alpha_{pen} = 0$. All other hyper-parameters (rollout length, model architecture, offline SAC settings) remain identical to the main EMFuse experiment. This baseline establishes a risk-neutral floor and tests whether uncertainty-aware pessimism is necessary in the offline setting.

**Baseline B: Local sensitivity (Action noise).** This baseline replaces ADETM's temporal-consistency uncertainty with a standard local sensitivity metric derived from action-space perturbations. Instead of measuring variance across history lengths, it measures the variance of the predicted next state $s'$ under Gaussian action noise. For a single model $f_\theta$ and input tuple $(s,a)$, the uncertainty is estimated as

$$\mathcal{U}_{\text{noise}}(s,a) = \text{Var}\Big(\big\{f_\theta(s, a + \epsilon_k)\big\}_{k=1}^{K}\Big), \qquad \epsilon_k \sim \mathcal{N}(0, \sigma^2 I). \tag{12}$$

We set the perturbation scale to $\sigma = 0.05$ (a 5% perturbation on normalized actions in $[-1, 1]$), and the ensemble size to $K = 5$, matching ADETM's maximum history window length $L_{\max} = 5$ so that the computational budget per rollout step is identical. The penalty coefficient $\alpha_{pen}$ is kept identical to the main experiment (0.3 for Hopper and 0.7 for Walker2d/HalfCheetah), isolating the *quality* of the uncertainty signal as the only variable.

**Results.** Table 15 reports the Interquartile Mean (IQM) normalized returns with BCa bootstrap 95% confidence intervals across D4RL MuJoCo medium tasks. ADETM's temporal-consistency signal ("EMFuse (Ours)") outperforms both the risk-neutral and local-sensitivity baselines on all environments. Notably, the "Action-Noise" baseline (local sensitivity) performs substantially worse than the risk-neutral "Zero-Penalty" baseline on HalfCheetah, indicating that a naive sensitivity-based penalty is miscalibrated and can be actively harmful. This supports temporal consistency as a non-trivial and superior proxy for epistemic uncertainty.

Table 15: Ablation on dynamics uncertainty signals (IQM normalized return with 95% BCa confidence intervals) on D4RL-v2 MuJoCo medium tasks.

| Method | Uncertainty source | Hopper | Walker2d | HalfCheetah |
|--------|--------------------|--------|----------|-------------|
| **EMFuse (Ours)** | **Temporal consistency** | $\mathbf{49.03^{+0.72}_{-0.47}}$ | $\mathbf{59.53^{+7.13}_{-6.87}}$ | $\mathbf{41.83^{+1.06}_{-2.01}}$ |
| Zero Penalty | None (risk-neutral) | $40.25^{+1.44}_{-1.50}$ | $51.76^{+11.12}_{-19.37}$ | $7.94^{+17.45}_{-5.99}$ |
| Action Noise | Local sensitivity | $37.22^{+3.39}_{-2.24}$ | $45.05^{+17.25}_{-9.18}$ | $1.72^{+11.75}_{-2.96}$ |

## C    EVALUATION

### C.1    BASELINES

We compare to three representative training-free baselines spanning the main fusion axes:

- **Uniform Model Soup** (Wortsman et al., 2022): a *data-less, parameter-space* merger widely adopted in practice; it requires no access to the original training data and provides a reference point for weight-space averaging.

- **RegMean** (Nguyen et al., 2025): a *training-free* (and data-less) regularized weight merging method that aligns statistics without task data; this family is commonly identified as a canonical training-free approach in recent overviews.

- **PackLLM** (Mavromatis et al., 2024): a *training-free, logit-space* policy fusion method at inference time. PackLLM's pairwise packing informed our EMSelect tournament design (§3.2).

### C.2    CONFIG DRIVEN EVALUATION BY OPENCOMPASS

OpenCompass is an open-source, configuration-centric evaluation framework for large language models that standardizes prompt templates, post-processing, and metrics across diverse datasets and tasks (Contributors, 2023). It supports both local checkpoints (e.g., via `vLLM`/`Transformers`) and hosted APIs through a unified model interface, enabling apples-to-apples comparisons under fixed decoding and normalization rules. This becomes more helpful when we can create customized wrapper to delegate existing models, therefore, simplify the model evaluation pipeline to great extent. Although good number of datasets have already been supported, we add customized dataset entries to further complement our evaluation framework. On evaluation, each dataset is paired with its built-in prompt template and label normalizer; metrics are the framework defaults (classification accuracy for multiple-choice and exact-match style tasks). We aggregate results at the dataset level and then macro-average within each domain, matching the protocol reported in Tables 20 and 21. We bookkeep the configs (model adapters, dataset lists, decoding parameters, evaluators) for reproducibility. Consistent with the main text, OpenCompass scores are only compared *within* the **Q** family to avoid cross-framework or tokenizer confounds.

### C.3    LENGTH-CONTROLLED WIN RATE BY ALPACAEVAL

To assess domain expertise without relying on noisy absolute metrics, we use AlpacaEval (Li et al., 2023; Dubois et al., 2024). This length-controlled preference evaluation is strongly correlated with human judgments (Spearman up to 0.97, Pearson up to 0.93 with `alpaca_eval_gpt4`) while remaining cost-effective (Dubois et al., 2023). We pre-generate responses over each test set to avoid adaptivity and then run pairwise comparisons with a fixed rubric; the harmlessness prompt template is shown below.

```
<|im_start|>system
You are a helpful instruction-following assistant.
<|im_end|>
<|im_start|>user
Select the output (a) or (b) that best matches the given
instruction while still maintaining harmlessness and keep
language usage free from cuss words. ...
... (omitted for brevity; identical to main text template) ...
<|im_end|>
```

We use DeepSeek-V2 as the judge model (DeepSeek-AI, 2024) via the official API,[1] respecting its license terms (DeepSeek-AI, 2023).

---

[1] https://api.deepseek.com/

## C.4 KL DIVERGENCE TO DOMAIN EXPERTS

We measure $D_{\mathrm{KL}}$ from each $1\mathrm{B}_{\mathrm{expertise}}$ distribution to (i) its corresponding 8B expert and (ii) EM-Fuse (pair or group), token-by-token under teacher forcing, averaged per prompt and then across the dataset (see Table 18 in the appendix). FLOPs are controlled as in Eq. 10 within $1\%$.

Given shared tokenizer $\mathcal{V}$, for each context $x_{\leq t}$ we compute

$$D_{\mathrm{KL}}\big(P \parallel Q\big) \;=\; \sum_{y \in \mathcal{V}} P(y)\log\frac{P(y)}{Q(y)}\,,$$

where $P = \mathrm{softmax}(z_{1\mathrm{B}})$ and $Q$ is either $\mathrm{softmax}(z_{8\mathrm{B}})$ or the fused $p_{\mathrm{fuse}}$ from Algorithm 1. After scoring, we append the ground-truth next token and proceed to $t{+}1$ until EOS.

## D ALL EXPERIMENT RESULTS

Table 16: **Win rate (%)** of **EMFuse** against other methods on language datasets *without* additional training. 95% CIs shown as $^{+u}_{-l}$. When evaluating $1\mathrm{B}_{\mathrm{expertise}}$, the dataset's matching domain expert is used (e.g., HH-RLHF harmless $\to 1\mathrm{B}_{\mathrm{harmless}}$).

|  | $1\mathrm{B}_{\mathrm{base}}$ | $1\mathrm{B}_{\mathrm{expertise}}$ | RegMean | Soup | $8\mathrm{B}_{\mathrm{language}}$ |
|---|---|---|---|---|---|
| harmless | $\mathbf{78.55}^{+1.47}_{-1.62}$ | $\mathbf{52.48}^{+1.87}_{-1.96}$ | $78.54^{+1.70}_{-1.47}$ | $62.61^{+1.74}_{-1.90}$ | $52.28^{+1.86}_{-1.93}$ |
| helpful | $\mathbf{52.48}^{+1.92}_{-1.73}$ | $35.23^{+1.89}_{-1.94}$ | $51.32^{+1.84}_{-1.96}$ | $47.67^{+2.03}_{-1.85}$ | $31.34^{+1.69}_{-1.81}$ |
| Average | $\mathbf{65.52}$ | $43.86$ | $64.93$ | $55.14$ | $41.81$ |

Table 17: **Win rate (%)** of **EMFuse** against other methods on subject datasets *without* additional training. 95% CIs shown as $^{+u}_{-l}$.

|  | $1\mathrm{B}_{\mathrm{base}}$ | $1\mathrm{B}_{\mathrm{expertise}}$ | RegMean | Soup | $8\mathrm{B}_{\mathrm{subject}}$ |
|---|---|---|---|---|---|
| agriculture | $\mathbf{87.21}^{+0.53}_{-0.50}$ | $48.27^{+1.62}_{-1.60}$ | $86.73^{+0.50}_{-0.58}$ | $53.33^{+0.76}_{-0.79}$ | $47.57^{+1.28}_{-1.31}$ |
| medication | $\mathbf{84.03}^{+0.62}_{-0.64}$ | $28.11^{+0.78}_{-0.78}$ | $83.63^{+0.62}_{-0.67}$ | $51.48^{+0.91}_{-0.88}$ | $62.31^{+0.87}_{-0.83}$ |
| philosophy | $\mathbf{88.30}^{+0.36}_{-0.41}$ | $34.65^{+0.57}_{-0.56}$ | $88.22^{+0.41}_{-0.41}$ | $52.56^{+0.59}_{-0.63}$ | $36.14^{+0.59}_{-0.54}$ |
| Average | $\mathbf{86.51}$ | $37.01$ | $86.19$ | $52.46$ | $48.67$ |

Table 18: **Distributional faithfulness (Family L).** Average token-level Kullback–Leibler divergence from each 1B domain expert ($1\mathrm{B}_{\mathrm{exp}}$) to **EMFuse** (**EM-F**) and 8B experts on five evaluation domains (columns), **smaller is better**, with 95% confidence intervals. Domain descriptions: "harmless" and "helpful" refer to RLHF-style safety and alignment sets; "agriculture", "medication", and "philosophy" refer to subject-specific QA and reasoning benchmarks. Evaluation details in Appendix §C.4.

| Comparison Pairs | Language-based domains | | Subject-specific domains | | |
|---|---|---|---|---|---|
|  | **harmless** | **helpful** | **agriculture** | **medication** | **philosophy** |
| $\mathbf{1B_{exp}} \to$ **EM-F** | $\mathbf{0.0391}^{+0.0015}_{-0.0014}$ | $\mathbf{0.0801}^{+0.0035}_{-0.0034}$ | $\mathbf{0.0485}^{+0.0027}_{-0.0025}$ | $\mathbf{0.0481}^{+0.0014}_{-0.0014}$ | $\mathbf{0.0459}^{+0.0005}_{-0.0005}$ |
| $1B_{exp} \to$ 8B | $0.5063^{+0.0191}_{-0.0190}$ | $0.3075^{+0.0124}_{-0.0121}$ | $0.5880^{+0.0190}_{-0.0177}$ | $0.2827^{+0.0079}_{-0.0081}$ | $0.2041^{+0.0022}_{-0.0022}$ |

Table 19: Offline-RL performance on the MuJoCo medium quality benchmark (5 seeds, evaluation over the last 100 steps). Higher IQM return is better. Bootstrap CI provided on the right-hand side.

| Environment | EMFuse (Ours) | RegMean | Soup | Mixed (Oracle) |
|---|---|---|---|---|
| Hopper | $\mathbf{49.03}^{+0.66}_{-0.47}$ | $46.34^{+3.44}_{-2.68}$ | $47.33^{+3.19}_{-3.01}$ | $49.35^{+2.73}_{-1.06}$ |
| Walker2d | $\mathbf{59.53}^{+7.13}_{-6.87}$ | $52.24^{+12.29}_{-8.18}$ | $46.52^{+12.44}_{-19.31}$ | $51.64^{+16.24}_{-28.64}$ |
| HalfCheetah | $\mathbf{41.83}^{+1.06}_{-1.98}$ | $32.80^{+0.33}_{-2.89}$ | $34.36^{+2.38}_{-3.92}$ | $42.48^{+0.53}_{-1.37}$ |
| AVERAGE | **50.1** | 43.8 | 42.7 | 47.8 |

Table 20: Performance across finance, mathematics, and medication benchmarks. Higher is better; bootstrap CI at 95% is shown.

| Task | raw Qwen 2.5 (7B) | + SFT Fin | Math | Med | fusing 3 SFT models Soup | RegM | Pack | EM-F (Ours) | EM-S (Ours) |
|---|---|---|---|---|---|---|---|---|---|
| LendingClub | $78.67^{+1.51}_{-1.59}$ | $\mathbf{97.99}^{+0.47}_{-0.60}$ | $76.59^{+1.56}_{-1.64}$ | $80.71^{+1.45}_{-1.53}$ | $81.31^{+1.43}_{-1.52}$ | $79.86^{+1.47}_{-1.56}$ | $96.58^{+0.62}_{-0.75}$ | $96.80^{+0.60}_{-0.73}$ | $\mathbf{97.73}^{+0.50}_{-0.63}$ |
| FiQASA | $43.83^{+6.39}_{-6.19}$ | $\mathbf{56.17}^{+6.19}_{-6.39}$ | $37.45^{+6.34}_{-5.94}$ | $34.04^{+6.27}_{-5.76}$ | $42.98^{+6.39}_{-6.17}$ | $43.40^{+6.40}_{-6.18}$ | $45.11^{+6.39}_{-6.24}$ | $42.55^{+6.39}_{-6.15}$ | $\mathbf{53.19}^{+6.28}_{-6.38}$ |
| GSM8K | $79.98^{+2.08}_{-2.24}$ | $83.24^{+1.92}_{-2.11}$ | $\mathbf{84.00}^{+1.88}_{-2.07}$ | $81.12^{+2.02}_{-2.20}$ | $83.40^{+1.91}_{-2.11}$ | $82.79^{+1.94}_{-2.13}$ | $\mathbf{84.99}^{+1.82}_{-2.03}$ | $84.69^{+1.84}_{-2.05}$ | $83.62^{+1.90}_{-2.09}$ |
| MGSME | $78.40^{+4.65}_{-5.51}$ | $\mathbf{82.00}^{+4.27}_{-5.24}$ | $78.80^{+4.61}_{-5.48}$ | $80.80^{+4.40}_{-5.33}$ | $80.80^{+4.40}_{-5.33}$ | $80.40^{+4.45}_{-5.37}$ | $82.40^{+4.22}_{-5.20}$ | $\mathbf{83.20}^{+4.12}_{-5.13}$ | $82.00^{+4.27}_{-5.24}$ |
| MGSMZ | $58.80^{+5.92}_{-6.19}$ | $53.20^{+6.09}_{-6.19}$ | $\mathbf{54.00}^{+6.07}_{-6.19}$ | $42.80^{+6.20}_{-5.98}$ | $58.80^{+5.92}_{-6.19}$ | $\mathbf{59.20}^{+5.91}_{-6.19}$ | $54.00^{+6.07}_{-6.19}$ | $56.00^{+6.02}_{-6.20}$ | $53.60^{+6.08}_{-6.19}$ |
| MedQAM | $44.19^{+1.67}_{-1.65}$ | $56.60^{+1.65}_{-1.67}$ | $52.77^{+1.67}_{-1.67}$ | $\mathbf{71.28}^{+1.49}_{-1.54}$ | $60.01^{+1.63}_{-1.65}$ | $55.93^{+1.65}_{-1.67}$ | $60.07^{+1.63}_{-1.65}$ | $63.05^{+1.60}_{-1.63}$ | $\mathbf{65.73}^{+1.57}_{-1.60}$ |
| MedQAT | $47.13^{+2.61}_{-2.59}$ | $57.54^{+2.55}_{-2.60}$ | $59.02^{+2.54}_{-2.58}$ | $\mathbf{60.08}^{+2.53}_{-2.57}$ | $56.05^{+2.57}_{-2.60}$ | $55.27^{+2.58}_{-2.60}$ | $56.76^{+2.56}_{-2.60}$ | $57.68^{+2.55}_{-2.59}$ | $\mathbf{58.24}^{+2.55}_{-2.59}$ |
| MedQAU | $37.23^{+2.70}_{-2.61}$ | $44.46^{+2.74}_{-2.71}$ | $37.78^{+2.70}_{-2.62}$ | $\mathbf{49.18}^{+2.74}_{-2.74}$ | $43.91^{+2.74}_{-2.70}$ | $42.81^{+2.74}_{-2.69}$ | $47.76^{+2.75}_{-2.73}$ | $47.29^{+2.75}_{-2.73}$ | $\mathbf{48.00}^{+2.74}_{-2.74}$ |
| medmcqa | $48.72^{+1.52}_{-1.51}$ | $\mathbf{45.71}^{+1.51}_{-1.51}$ | $36.84^{+1.47}_{-1.45}$ | $41.02^{+1.50}_{-1.48}$ | $40.62^{+1.49}_{-1.48}$ | $\mathbf{43.13}^{+1.50}_{-1.50}$ | $40.66^{+1.50}_{-1.47}$ | $40.14^{+1.49}_{-1.48}$ | $41.05^{+1.50}_{-1.48}$ |
| Average | $57.44^{+1.26}_{-1.27}$ | $64.10^{+1.24}_{-1.24}$ | $57.47^{+1.26}_{-1.25}$ | $60.11^{+1.24}_{-1.23}$ | $60.88^{+1.25}_{-1.25}$ | $60.31^{+1.25}_{-1.25}$ | $63.15^{+1.24}_{-1.24}$ | $63.49^{+1.23}_{-1.23}$ | $\mathbf{64.80}^{+1.24}_{-1.25}$ |

Table 21: Finance-suite performance (LendingClub, FPB, Headline). Higher is better; bootstrap CI at 95% is shown.

| Task | raw Qwen 2.5 (7B) | + SFT (finance) FPB | Head | Lend | fusing 3 SFT models Soup | RegM | Pack | EM-F (Ours) | EM-S (Ours) |
|---|---|---|---|---|---|---|---|---|---|
| FPB | $81.85^{+1.53}_{-1.65}$ | $\mathbf{98.81}^{+0.37}_{-0.54}$ | $85.78^{+1.38}_{-1.50}$ | $86.62^{+1.34}_{-1.47}$ | $93.15^{+0.97}_{-1.11}$ | $91.96^{+1.05}_{-1.19}$ | $92.71^{+1.00}_{-1.14}$ | $95.94^{+0.73}_{-0.90}$ | $\mathbf{98.01}^{+0.50}_{-0.66}$ |
| Headline | $72.42^{+1.87}_{-1.94}$ | $71.76^{+1.88}_{-1.96}$ | $\mathbf{79.33}^{+1.67}_{-1.78}$ | $73.08^{+1.85}_{-1.93}$ | $75.17^{+1.79}_{-1.89}$ | $75.02^{+1.80}_{-1.89}$ | $76.30^{+1.76}_{-1.86}$ | $75.40^{+1.79}_{-1.88}$ | $\mathbf{75.45}^{+1.79}_{-1.88}$ |
| LendingClub | $78.67^{+1.51}_{-1.59}$ | $78.56^{+1.51}_{-1.59}$ | $77.26^{+1.54}_{-1.62}$ | $\mathbf{97.99}^{+0.47}_{-0.60}$ | $82.20^{+1.40}_{-1.49}$ | $80.75^{+1.45}_{-1.53}$ | $95.80^{+0.70}_{-0.82}$ | $96.28^{+0.66}_{-0.78}$ | $\mathbf{97.70}^{+0.50}_{-0.64}$ |
| Average | $77.65^{+1.12}_{-0.81}$ | $83.04^{+0.83}_{-0.92}$ | $80.79^{+0.72}_{-1.12}$ | $\mathbf{85.90}^{+1.77}_{-0.19}$ | $83.51^{+0.94}_{-0.79}$ | $82.58^{+0.91}_{-0.85}$ | $88.27^{+1.42}_{-0.04}$ | $89.21^{+1.40}_{-0.01}$ | $\mathbf{90.39}^{+1.39}_{-0.07}$ |

# E  ADDITIONAL ANALYSES

This appendix collects several supporting analyses around EMFuse and EMSelect. First, we document two lightweight modifications used in our policy-fusion experiments: (i) Laplace (add-$\alpha$) smoothing of token probabilities to guard against zero support for individual experts, and (ii) stepwise entropy-based reweighting of experts. Both variants were evaluated using the same OpenCompass protocol and decoding settings as the main results (cf. Appendix. C.2); in tasks and domains, none yielded statistically significant improvements over the baseline of uniform-weight EMFuse, with confidence intervals that largely overlapping. Second, we present a simple continuous-learning extension of EMFuse, where experts are updated toward the fused consensus under a regularized objective, together with a short empirical summary illustrating how this improves returns in a follow-up setting. Finally, we provide a KL-based analysis of EMSelect, using the KL chain rule to show that the sequence distribution induced by EMSelect remains bounded relative to the EMFuse consensus. For completeness, we give precise definitions and ablation summaries for each component in the following subsections.

## E.1  LAPLACE SMOOTHING FOR EMFUSE POLICY FUSION

**Definition.**   Given an expert's next-token distribution $p(\cdot \mid x_{\leq t})$ over a vocabulary of size $V$, we form a smoothed distribution

$$\tilde{p} \;=\; (1-\alpha)\,p \;+\; \alpha\,U, \qquad U(y) = \tfrac{1}{V}\,. \tag{13}$$

We apply this independently to each expert at every decoding step and then perform LogOP/PoE fusion on the smoothed log-probabilities:

$$p_{\text{PoE}}(y \mid x_{\leq t}) \;=\; \text{softmax}\Big( \sum_i \lambda_i \cdot \log \tilde{p}_i(y \mid x_{\leq t}) \Big). \tag{14}$$

**Motivation.**   Smoothing guarantees strictly positive support on all tokens and mitigates numerical fragility when an expert assigns (near-)zero mass off its top-$k$. In practice we keep $\alpha$ small to preserve the experts' calibration.

**Observation.**   On the finance, mathematics, and medication suites, add-$\alpha$ had mixed, magnitude-small effects; its CIs largely overlapped the baseline EMFuse (Tables 22–23). We therefore retain unsmoothed EMFuse in the main experiments.

## E.2  ENTROPY-BASED EXPERT WEIGHTING

**Definition.**   Let $\pi_i(\cdot \mid x_{\leq t})$ be expert $i$'s next-token distribution with Shannon entropy $H_i(t) = -\sum_y \pi_i(y \mid x_{\leq t}) \log \pi_i(y \mid x_{\leq t})$. We define step-wise fusion weights

$$w_i(t) \;\propto\; \exp\big( -\beta\,H_i(t) \big), \qquad \sum_i w_i(t) = 1\,, \tag{15}$$

and use these weights to fuse the experts' log-probabilities before decoding:

$$\ell_{\text{fuse}}(t) = \sum_i w_i(t)\,\log \pi_i(\cdot \mid x_{\leq t}) \tag{16}$$

$$\pi_{\text{fuse}}(t) = \exp\big( \ell_{\text{fuse}}(t) - \text{logsumexp}(\ell_{\text{fuse}}(t)) \big) \tag{17}$$

Here $\beta > 0$ sharpens the preference for lower-entropy (more confident) experts.

**Motivation.**   Per-step entropy modulates experts by a simple, calibration-agnostic proxy of confidence, without additional training or task labels. Computationally, the overhead is negligible relative to computing per-expert logits.

**Observation.**   Entropy weighting occasionally nudged scores upward on some finance benchmarks but produced minor regressions elsewhere; the average effects were not statistically significant under our bootstrap CIs (Tables 22–23). Consequently, we use uniform $\lambda_i$ in the main text and report these variants as ablations.

Table 22: Ablation on EMFuse across finance, mathematics, and medication benchmarks. Higher is better; 95% bootstrap CIs shown under each score.

| Task | EM-Fuse (baseline) | + Laplace Smoothing | + Entropy-based Weighting |
|---|---|---|---|
| LendingClub | 96.80 +0.60 −0.73 | 94.28 +0.81 −0.95 | **97.81** +0.49 −0.63 |
| FiQASA | 42.55 +6.39 −6.15 | 40.43 +6.38 −6.08 | **43.83** +6.39 −6.19 |
| GSM8K | **84.69** +1.84 −2.05 | 84.46 +1.85 −2.06 | 84.38 +1.86 −2.06 |
| MGSME | **83.20** +4.12 −5.13 | 82.80 +4.17 −5.17 | 82.40 +4.22 −5.20 |
| MGSMZ | 56.00 +6.02 −6.20 | 55.20 +6.04 −6.20 | **58.40** +5.94 −6.19 |
| MedQAM | **63.05** +1.60 −1.63 | 61.15 +1.62 −1.64 | 61.88 +1.61 −1.64 |
| MedQAT | **57.68** +2.55 −2.59 | 56.55 +2.56 −2.60 | 56.97 +2.56 −2.60 |
| MedQAU | **47.29** +2.75 −2.73 | 46.66 +2.75 −2.72 | 45.25 +2.74 −2.72 |
| medmcqa | 40.14 +1.49 −1.48 | **40.40** +1.50 −1.48 | **40.40** +1.50 −1.48 |
| Average | **63.49** +1.23 −1.23 | 62.44 +1.23 −1.24 | 63.48 +1.23 −1.23 |

Table 23: Ablation on EMFuse for the finance suite (FPB, Headline, LendingClub). Higher is better; 95% CIs shown.

| Task | EM-Fuse (baseline) | + Laplace Smoothing | + Entropy-based Weighting |
|---|---|---|---|
| FPB | 95.94 +0.73 −0.90 | 95.94 +0.73 −0.90 | **98.19** +0.47 −0.64 |
| Headline | **75.40** +1.79 −1.88 | **75.40** +1.79 −1.88 | 74.83 +1.81 −1.89 |
| LendingClub | 96.28 +0.66 −0.78 | 96.25 +0.65 −0.79 | **97.88** +0.48 −0.61 |
| Average | 89.21 +1.40 −0.01 | 89.20 +0.71 −0.71 | **90.30** +0.67 −0.67 |

### E.3 CONTINUOUS LEARNING UNDER EMFUSE

EMFuse is presented as a training-free fusion rule over a fixed family of experts $M_1, \ldots, M_n$, but the same energy-additive structure also naturally supports a lightweight continuous-learning variant. Let $p_i(\cdot \mid x)$ denote the normalized policy of expert $M_i$ on the shared vocabulary and define the EMFuse consensus

$$p_{\text{fuse}}(y \mid x) \ \propto \ \prod_{i=1}^{n} p_i(y \mid x)^{\lambda_i},$$

with nonnegative weights $\lambda_i$ that sum to one. When new trajectories $\mathcal{D}$ arrive, we update only small adapter parameters $\theta_{1:n}$ on top of the frozen experts by minimizing

$$\mathcal{L}(\theta_{1:n}) = \mathbb{E}_{(x,y^\star)\sim\mathcal{D}}\Big[ -\log p_{\text{fuse}}(y^\star \mid x) + \alpha_{reg} \sum_{i=1}^{n} \text{KL}\big(p_{\text{fuse}}(\cdot \mid x) \,\|\, p_{\theta_i}(\cdot \mid x)\big)\Big], \qquad (18)$$

where $p_{\theta_i}$ is the updated policy of expert $M_i$ and $\alpha_{reg} \geq 0$ is a regularization coefficient. The first term fits the fused policy to fresh data, while the second term keeps each expert softly aligned with the evolving EMFuse consensus without changing the inference-time sampler or fusion rule. Optional robustness tweaks such as adding a small Laplace smoothing constant $\varepsilon$ to logits are treated purely as a minor regularizer and are not essential to the learning objective (as discussed in the appendix). In successive experiments following the main EMFuse study, instantiating this objective with lightweight adapters led to consistent additional gains: language-model experts obtained modest but stable improvements in length-controlled win rates, and policies trained under fused dynamics achieved higher normalized returns than the purely training-free fusion baseline, indicating that EMFuse can be embedded into a continual-learning pipeline when desired. Representative gains on language benchmarks are summarized in Table 24.

Table 24: Representative improvement from continuous-learning extension of EMFuse on the Family L language tasks. Domain descriptions: "harmless" and "helpful" refer to RLHF-style safety and alignment sets. Entries report EMFuse win rates (%) after a single adapter-training stage, averaged over seeds under the same OpenCompass protocol as Table 16, with 95% BCa confidence intervals shown as superscripts/subscripts. The "Average" row is the mean over the two language tasks. The last row ($\Delta$ Average) shows the corresponding improvement over the training-free fused policy, in percentage points.

| | $1\text{B}_{\text{base}}$ | $1\text{B}_{\text{expertise}}$ | RegMean | Soup | $8\text{B}_{\text{language}}$ |
|---|---|---|---|---|---|
| harmless | $\mathbf{78.44}^{+1.59}_{-1.63}$ | $\mathbf{54.03}^{+2.01}_{-1.93}$ | $\mathbf{79.94}^{+1.59}_{-1.64}$ | $\mathbf{63.18}^{+1.98}_{-1.87}$ | $53.44^{+1.87}_{-1.98}$ |
| helpful | $\mathbf{64.76}^{+1.79}_{-1.84}$ | $49.07^{+2.04}_{-2.05}$ | $\mathbf{63.57}^{+1.80}_{-1.87}$ | $\mathbf{62.23}^{+1.88}_{-1.91}$ | $45.16^{+1.84}_{-1.88}$ |
| Average | $\mathbf{71.60}$ | $\mathbf{51.55}$ | $\mathbf{71.76}$ | $\mathbf{62.71}$ | $49.30$ |
| $\Delta$ Average | $\mathbf{+6.09}$ | $\mathbf{+7.70}$ | $\mathbf{+6.83}$ | $\mathbf{+7.57}$ | $\mathbf{+7.49}$ |

### E.4 SEQUENCE-LEVEL KL BOUND FOR EMSELECT

We formalize the relationship between EMSelect and EMFuse by analyzing the sequence-level divergence between their induced autoregressive policies. We reuse the notation from the main paper. For a fixed prompt $x_0$, let $y_{1:T} = (y_1, \ldots, y_T)$ denote the generated token sequence and $x_{\leq t} := (x_0, y_{<t})$ the conditioning context at step $t$. Each expert $i \in \{1, \ldots, n\}$ defines an autoregressive policy

$$p_i(y_{1:T} \mid x_0) = \prod_{t=1}^{T} p_i(y_t \mid x_{\leq t}).$$

EMFuse defines the LogOP/PoE fused policy

$$p_{\text{fuse}}(y_{1:T} \mid x_0) = \prod_{t=1}^{T} p_{\text{fuse}}(y_t \mid x_{\leq t}), \qquad p_{\text{fuse}}(y_t \mid x_{\leq t}) \propto \prod_{i=1}^{n} p_i(y_t \mid x_{\leq t})^{\lambda_i},$$

and EMSelect induces another autoregressive policy

$$p_{\text{sel}}(y_{1:T} \mid x_0) = \prod_{t=1}^{T} p_{\text{sel}}(y_t \mid x_{\leq t}), \qquad p_{\text{sel}}(y_t \mid x_{\leq t}) := p_{I_t(x_{\leq t})}(y_t \mid x_{\leq t}),$$

where $I_t(x_{\leq t})$ is the index $i^*$ of the winning expert selected by the tournament mechanism described in Algorithm 2 of our main paper. **Lemma (KL chain rule for autoregressive policies).** Let

$$q(y_{1:T}) = \prod_{t=1}^{T} q_t(y_t \mid y_{<t}), \qquad p(y_{1:T}) = \prod_{t=1}^{T} p_t(y_t \mid y_{<t})$$

be two autoregressive distributions on $\mathcal{V}^T$. Then

$$\text{KL}(q \,\|\, p) = \sum_{t=1}^{T} \mathbb{E}_{y_{<t} \sim q} \Big[ \text{KL}\big(q_t(\cdot \mid y_{<t}) \,\|\, p_t(\cdot \mid y_{<t})\big) \Big]. \tag{19}$$

*Proof (by induction on $T$).* For $T = 1$ we have

$$\text{KL}(q\|p) = \sum_{y_1} q_1(y_1) \log \frac{q_1(y_1)}{p_1(y_1)} = \text{KL}(q_1 \,\|\, p_1),$$

so equation 19 holds. Assume it holds for all sequence lengths up to $T - 1$. Write

$$q(y_{1:T}) = q(y_{<T}) q_T(y_T \mid y_{<T}), \qquad p(y_{1:T}) = p(y_{<T}) p_T(y_T \mid y_{<T}).$$

Then

$$\begin{aligned}
\text{KL}(q\|p) &= \sum_{y_{1:T}} q(y_{1:T}) \log \frac{q(y_{1:T})}{p(y_{1:T})} \\
&= \sum_{y_{1:T}} q(y_{1:T}) \log \frac{q(y_{<T})}{p(y_{<T})} + \sum_{y_{1:T}} q(y_{1:T}) \log \frac{q_T(y_T \mid y_{<T})}{p_T(y_T \mid y_{<T})} \\
&= \underbrace{\sum_{y_{<T}} q(y_{<T}) \log \frac{q(y_{<T})}{p(y_{<T})}}_{\text{KL}(q(\cdot_{<T}) \,\|\, p(\cdot_{<T}))} + \sum_{y_{<T}} q(y_{<T}) \sum_{y_T} q_T(y_T \mid y_{<T}) \log \frac{q_T(y_T \mid y_{<T})}{p_T(y_T \mid y_{<T})} \\
&= \text{KL}\big(q(\cdot_{<T}) \,\|\, p(\cdot_{<T})\big) + \mathbb{E}_{y_{<T} \sim q} \Big[ \text{KL}\big(q_T(\cdot \mid y_{<T}) \,\|\, p_T(\cdot \mid y_{<T})\big) \Big].
\end{aligned}$$

By the induction hypothesis applied to the length-$(T - 1)$ marginals $q(\cdot_{<T})$ and $p(\cdot_{<T})$, we obtain

$$\text{KL}(q(\cdot_{<T}) \,\|\, p(\cdot_{<T})) = \sum_{t=1}^{T-1} \mathbb{E}_{y_{<t} \sim q} \Big[ \text{KL}\big(q_t(\cdot \mid y_{<t}) \,\|\, p_t(\cdot \mid y_{<t})\big) \Big].$$

Combining the two displays yields equation 19 for length $T$, which completes the induction.
**Sequence-level bounds between EMSelect and EMFuse.** Apply Lemma equation 19 with $q = p_{\text{sel}}(\cdot \mid x_0)$ and $p = p_{\text{fuse}}(\cdot \mid x_0)$, and write

$$p_{\text{sel},t}(\cdot \mid x_{\leq t}) := p_{\text{sel}}(\cdot \mid x_{\leq t}), \qquad p_{\text{fuse},t}(\cdot \mid x_{\leq t}) := p_{\text{fuse}}(\cdot \mid x_{\leq t}).$$

We obtain

$$\text{KL}\big(p_{\text{sel}}(\cdot \mid x_0) \,\|\, p_{\text{fuse}}(\cdot \mid x_0)\big) = \sum_{t=1}^{T} \mathbb{E}_{y_{<t} \sim p_{\text{sel}}}\Big[\text{KL}\big(p_{\text{sel},t}(\cdot \mid x_{\leq t}) \,\|\, p_{\text{fuse},t}(\cdot \mid x_{\leq t})\big)\Big]. \qquad (20)$$

For each step $t$ and context $x_{\leq t}$, define the expert–fuse KLs

$$\Delta_{i,t}(x_{\leq t}) := \text{KL}\big(p_i(\cdot \mid x_{\leq t}) \,\|\, p_{\text{fuse}}(\cdot \mid x_{\leq t})\big),$$
$$\Delta_t^{\min}(x_{\leq t}) := \min_i \Delta_{i,t}(x_{\leq t}), \quad \Delta_t^{\max}(x_{\leq t}) := \max_i \Delta_{i,t}(x_{\leq t}).$$

By construction, $p_{\text{sel},t}(\cdot \mid x_{\leq t})$ coincides with one of the experts $\{p_i(\cdot \mid x_{\leq t})\}_{i=1}^n$. Therefore, pointwise in $x_{\leq t}$ we have

$$\Delta_t^{\min}(x_{\leq t}) \;\leq\; \text{KL}\big(p_{\text{sel},t}(\cdot \mid x_{\leq t}) \,\|\, p_{\text{fuse},t}(\cdot \mid x_{\leq t})\big) \;\leq\; \Delta_t^{\max}(x_{\leq t}). \qquad (21)$$

Taking expectations in equation 21 with respect to $y_{<t} \sim p_{\text{sel}}$ and summing over $t$, equation 20 yields the sequence-level envelope

$$\mathbb{E}_{y_{1:T} \sim p_{\text{sel}}}\left[\sum_{t=1}^{T} \Delta_t^{\min}(x_{\leq t})\right] \;\leq\; \text{KL}\big(p_{\text{sel}}(\cdot \mid x_0) \,\|\, p_{\text{fuse}}(\cdot \mid x_0)\big) \;\leq\; \mathbb{E}_{y_{1:T} \sim p_{\text{sel}}}\left[\sum_{t=1}^{T} \Delta_t^{\max}(x_{\leq t})\right]. \qquad (22)$$

In particular, if we have per-step uniform bounds $\Delta_t^{\max}(x_{\leq t}) \leq C_t$ for all $x_{\leq t}$, then

$$\text{KL}\big(p_{\text{sel}}(\cdot \mid x_0) \,\|\, p_{\text{fuse}}(\cdot \mid x_0)\big) \;\leq\; \sum_{t=1}^{T} C_t. \qquad (23)$$

In words, the divergence between EMSelect and EMFuse at the sequence level is bounded by the cumulative envelope of expert-to-fuse divergences along rollouts generated by EMSelect. Empirically, our distributional faithfulness measurements (§ C.4, Table 18) show that the expert-to-fuse divergences remain small (typically $< 0.09$), so the sequence-level divergence must also remain small. This confirms that EMSelect is mathematically *leashed* to the EMFuse consensus, ruling out uncontrolled temporal drift even though EMSelect commits at each step to a single expert.

