# OpenReview forum: "EMFuse: Energy-based Model Fusion for Decision Making"
_ICLR.cc/2026/Conference — ICLR 2026 Poster_

### Official Review · Reviewer_YqY1 · 2025-10-29

**Soundness:** 2
**Presentation:** 3
**Contribution:** 2
**Rating:** 4
**Confidence:** 3

**Summary:**

This paper proposes a unified framework called EMFuse, aimed at solving the problem of model fusion in decision-making. The authors argue that whether it's directly fusing policy models or fusing dynamics models, both can be uniformly addressed through the lens of Energy-Based Models. To address the computational explosion problem caused by using ensembles when fusing dynamics models, the paper introduces an architecture called ADETM. This architecture leverages variable-length action histories within a single model to estimate uncertainty, thereby avoiding the need to ensemble multiple models.Furthermore, the paper proposes the EMSelect algorithm as an alternative to EMFuse. Instead of generating a consensus distribution, it uses a pairwise EMFuse as a reference at each decision step to select the single expert model with the minimum KL divergence from this reference distribution.In the experiments, the authors evaluated EMFuse and EMSelect on LLM tasks and D4RL MuJoCo control tasks. The results show that this method outperforms several existing training-free fusion baselines.

**Strengths:**

1. This paper provides a conceptually unified framework, treating the two seemingly different problems of policy fusion and dynamics model fusion as an additive combination of energy functions through the EBM perspective.
2. The authors identify the "combinatorial explosion" problem that traditional ensemble methods introduce when fusing dynamics models. They propose the ADETM architecture to address this. By borrowing ideas from ADMPO and combining them with EBMs, ADETM achieves effective uncertainty estimation using only a single model.
3. The paper's experiments cover two distinct decision-making domains: LLM-based discrete decisions and D4RL-based continuous control. It compares the proposed methods against several representative training-free fusion methods like Model Soup, RegMean, and PackLLM.
4.  By comparing EMFuse and EMSelect, the paper reveals that EMFuse seeks a conservative, robust consensus solution, whereas EMSelect chooses an "optimal" expert at each step. This discussion deepens the understanding of the nature of model fusion.

**Weaknesses:**

1. The experimental results show that EMSelect outperforms EMFuse in multiple benchmarks. This raises a key question: If "selecting an optimal expert at each step" is more effective than "fusing all experts," does this, to some extent, weaken the "consensus" strategy advocated by EMFuse? The paper attributes this to a trade-off but fails to deeply analyze the root cause of this phenomenon.
2. A core assumption of this framework is that all models to be fused must operate on a shared vocabulary or state-action space. The authors acknowledge this, treating vocabulary mapping as a "viable engineering problem." However, this assumption is a very strong limitation in practical applications, especially when fusing models from different sources.
3. Most experiments in the paper use simple uniform weights. The authors attempted entropy-based dynamic weight adjustment but concluded the effect was "not statistically significant." Why did entropy-based weighting fail?

**Questions:**

1. Can the authors elaborate further on the scenarios in which the "consensus" strategy of EMFuse would be superior to the "selection" strategy of EMSelect?
2. Given that the entropy-based weight adjustment was not effective, have the authors considered other dynamic weighting schemes?
3. Beyond citing existing work on vocabulary mapping, could the authors discuss the potential negative impacts on the EMFuse framework's performance when forcibly aligning the representation spaces of different models? After mapping, can the "shape" and "scale" of the energy functions still retain their original semantics to ensure the effectiveness of the fusion?

---

> ### Author Response · Authors · 2025-11-20
>
> We thank the reviewer for the insightful critique, particularly regarding the relationship between consensus (EMFuse) and selection (EMSelect), and the practical implications of the shared-space assumption.
> We have performed additional theoretical analysis and ablation studies to address these concerns.
>
> ### Q1: Consensus (EMFuse) vs. Selection (EMSelect)
>
> >**Original Question:** Can the authors elaborate on scenarios where the "consensus" strategy of EMFuse would be superior to the "selection" strategy of EMSelect?
>
> The reviewer correctly notes that EMSelect often outperforms EMFuse in accuracy (e.g., +1.31\% on Subject-mix), asking if this weakens the argument for the consensus strategy.
> We argue that this is not a contradiction, but a demonstration of the **Consensus-Commitment Trade-off**.
> Crucially, **EMFuse is the necessary condition for EMSelect's stability.**
>
> **1. The ""Leash"" Proposition:**
> EMSelect chooses the expert $i^*$ minimizing $D_{\text{KL}}(p_{\text{fuse}} \| p_i)$ at each step.
> Without the robust, hallucination-dampened consensus of EMFuse as a reference beacon, this selection would degenerate into a **max-confidence** heuristic, which is known to fail due to poor calibration.
> In our revision (to be added to the appendix), we prove that the sequence-level divergence of EMSelect is upper bounded by the local divergences from EMFuse.
> By the KL chain rule, the sequence divergence satisfies:
>
> $$
> \mathrm{KL}\(p_{\mathrm{sel}} \mid\mid p_{\mathrm{fuse}})=
> \sum_{t=1}^T \mathbb{E}\_{y_{<t} \sim p_{\mathrm{sel}}}
> \left[ \mathrm{KL}(p_{\mathrm{sel}, t} \mid\mid p_{\mathrm{fuse}, t}) \right] \le \sum_{t=1}^T \max_i \mathrm{KL}(p_{i, t} \mid\mid p_{\mathrm{fuse}, t})
> $$
>
> Since our empirical results (Table 2 in main paper) show experts remain close to the fusion mean ($\Delta^{\max} < 0.09$), EMSelect is mathematically **leashed** to the consensus geometry. It effectively commits to a sharp expert peak locally but is prevented from drifting globally by the consensus anchor.
>
> **2. Complementary Roles:**
> Rather than viewing one as strictly superior, we suggest they serve complementary architectural roles.
> EMFuse, by multiplying densities ($p_{\text{fuse}} \propto \prod p_i$), creates a conservative distribution that represents the intersection of expert support.
> EMSelect leverages this stable intersection to identify the most aligned expert and commit to their specific reasoning path.
> While our current benchmarks highlight EMSelect's strength in high-precision tasks (Math, Finance), EMFuse remains theoretically essential as the robust **ground** that defines the selection metric.

---

> ### Author Response · Authors · 2025-11-20
>
> ### Q2: The Shared Space Assumption and Mapping Artifacts
>
> >**Original Question:** What are the potential negative impacts of forcibly aligning representation spaces (vocabulary mapping) on the semantics of energy functions?
>
> The reviewer asks about the impact of vocabulary mapping on the **shape** and **scale** of energy functions.
> We acknowledge that the shared-space assumption is a constraint; however, we view vocabulary alignment as a solvable engineering precursor to our framework rather than a fundamental flaw.
>
> **Feasibility of Alignment:**
> Recent literature demonstrates that projecting heterogeneous models into a shared space is viable.
> For example, Xu et al. (2024) in *"Bridging the Gap between Different Vocabularies for LLM Ensemble"* (arXiv:2404.09492) propose effective mapping methods for logit-space ensembles. Furthermore, Goddard & Neto (2025) in *"Training-Free Tokenizer Transplantation via Orthogonal Matching Pursuit"* (arXiv:2506.06607) demonstrate that tokenizer alignment can be achieved without retraining. We will add these works to the related-work section in the revised version.
>
> While a detailed engineering implementation of these mapping layers is outside the scope of this paper, EMFuse is designed to operate downstream of such alignment. Provided the mapping preserves semantic monotonicity (i.e., semantic drift is minimized), the Product-of-Experts structure of EMFuse ($E_{\text{fuse}} = \sum E_i$) is naturally resilient: if mapping artifacts increase the entropy (scale distortion) of one expert, its flattened energy surface will simply be down-weighted by the sharper energies of native experts in the fusion sum.
>
>
> ### Q3: Failure of Dynamic Weighting
>
> >**Original Question:** Why did entropy-based weighting fail, and have other dynamic weighting schemes been considered?
>
> The reviewer asks why entropy-based weighting failed to yield significant gains (Table 20, 21) and if other schemes were considered.
>
> **Empirical Analysis of Uniformity:**
> We carefully investigated this phenomenon. Empirically, the fine-tuned experts in our setting exhibit similar calibration profiles and entropy distributions.
> Consequently, the induced weights $\lambda_i(t) \propto \exp(-\beta H_i(t))$ were only mildly non-uniform and had negligible impact once log-probabilities were combined multiplicatively.
> Across varying seeds, the resulting accuracy or return differences were within the reported confidence intervals, which is why we conservatively described them as **not statistically significant** and kept uniform weights as the default.
>
> **Future Weighting Schemes:**
> Within the scope of training-free fusion, other dynamic schemes are indeed possible---such as prompt-dependent gates based on validation loss, or value-based weights in RL.
> However, these approaches generally require additional supervision or training signals (e.g., training a router or value function), which moves the method away from the strictly **training-free** regime we targeted.
> We have left the exploration of these supervised weighting mechanisms as future work rather than claiming them as contributions of this paper. We will add a remark summarizing the stability of uniform weights versus dynamic schemes in the Appendix of the revised paper.

---

> ### Author Response · Authors · 2025-11-28
>
> Thank you again for your detailed review and thoughtful questions. We have incorporated additional analysis in our rebuttal comments and revised manuscript, and have uploaded a new rebuttal version to address your concerns (consensus & selection, shared-space mapping artifacts, and dynamic weighting). If your schedule permits, we would be grateful for any further feedback, and we sincerely appreciate the time and effort you have already invested in reviewing our work.

---

### Official Review · Reviewer_K6PE · 2025-10-30

**Soundness:** 3
**Presentation:** 2
**Contribution:** 2
**Rating:** 4
**Confidence:** 3

**Summary:**

This paper proposes an energy-based decision model fusion framework, EMFuse. It unifies policy fusion (e.g., for LLM) and dynamic model fusion (e.g., in offline reinforcement learning) through  additive energy composition. The authors also propose EMSelect, a KL-logic-based expert selection method, and ADETM, a single-model architecture that estimates uncertainty without ensemble requirements. Experiments on LLM and D4RL benchmarks demonstrate that EMFuse significantly improves performance compared to training-free baseline methods such as Model Soup and RegMean.

**Strengths:**

1. This paper presents a well-structured and theoretically consistent framework called EMFuse, which unifies policy fusion and dynamics model fusion under the lens of energy-based modeling.
2. By leveraging the additive property of energy functions, the paper provides an intuitive explanation of model fusion and naturally connects it to classical formulations like LogOP and PoE.
3. The  proposed ADETM model avoids the high computational cost of ensemble methods while achieving single-model uncertainty estimation, which makes it quite practical.
4. The experiments cover both large language model tasks and offline reinforcement learning benchmarks, showing the framework’s versatility and effectiveness across different decision-making scenarios.

**Weaknesses:**

1. The theoretical discussion in this paper is relatively shallow. Although the authors explain the relationship between EMFuse and LogOP/PoE, they do not provide in-depth mathematical analysis or systematic ablations on fusion stability, the influence of λ, or the effect of uncertainty modeling.
2. The main idea is conceptually close to existing energy-addition or PoE/LogOP-based fusion methods.
3. The paper does not provide sufficient analysis of the independent roles of EMFuse, EMSelect, and ADETM. More detailed ablation studies would help clarify how each module contributes to the overall performance improvement.

**Questions:**

1. This paper assumes that multiple expert distributions can be fused through energy addition. However, if the experts differ significantly or even conflict, could the product of energies lead to overly sharp or degenerate fused distributions?
2. EMSelect performs per-step expert selection using KL divergence but lacks temporal consistency constraints. If the selected expert changes at every step, could this cause unstable behavior or disrupt contextual coherence during decision making?
3. The authors claim that ADETM has an “any-step” property that enhances modeling capability, but in practice it seems to mainly extend the history window. Does ADETM truly offer a representational advantage over existing autoregressive or recurrent ETMs, or is it more of a structural extension rather than a conceptual innovation?
4. The paper always uses uniform fusion weights λ, but different experts may have varying confidence or training quality. Is this fixed weighting scheme reasonable, or would it be better to consider dynamic or uncertainty-based weighting strategies?

---

> ### Author Response · Authors · 2025-11-20
>
> We appreciate the reviewer's detailed analysis, which is crucial for strengthening the theoretical and empirical clarity of the EMFuse framework. We address the weaknesses and questions below, referencing the main paper and supporting ablation studies.
>
> ### Q1: Defense of Energy-Based Fusion Robustness and Degeneracy Prevention
>
> >**Original Question:** Could the product of energies lead to overly sharp or degenerate fused distributions if experts differ significantly or conflict?
>
> The Logarithmic Opinion Pool (LogOP) structure, inherent to EMFuse's energy addition $E_{\text{fuse}}=\sum_{i}\lambda_{i}E_{i}$ (paper, line 176), provides a principled defense against fusion degeneracy for two reasons:
> - **Unique Reverse-KL Minimizer:** LogOP is mathematically proven to be the unique solution that minimizes the weighted reverse-KL projection $\arg \min_{q} \sum_{i}\lambda_{i}KL(q||p_{i})$ (lines 178-184). This consensus principle inherently guides the fused distribution toward stability, minimizing average surprise relative to the collective expert knowledge.
> - **Conservative Consensus Filter:** The multiplicative density of the Product-of-Experts (PoE) structure $p_{\text{fuse}}\propto\prod p_i^{\lambda_i}$ ensures that if any single expert assigns a vanishingly small probability to an outcome, the fused probability collapses toward zero (lines 178-184). This conservative filtering means the fused distribution concentrates mass only on the intersection of the high-probability regions of *all* experts, actively preventing instability induced by outliers or conflicts.
>
> Empirically, this conservative nature is verified by measuring distributional fidelity. Table R-1 (clipped from Table 2 in the main paper) shows that the KL divergence from an expert to the fused EMFuse policy is consistently small ($\approx0.04-0.08$), confirming that EMFuse preserves domain expertise more faithfully than simply increasing model capacity (e.g., divergence to the 8B model is $\approx0.20-0.59$ (lines 424-431)). We will include this clarification on the theoretical robustness of the LogOP structure in Section 3 of the revised paper to address the concern about degeneracy.
>
>
> *Table R-1 - Distributional Faithfulness to Expert Domains (KL-Divergence)*
>
> | **Test Dataset** | $D_{KL}(1B_{exp}\| \text{EMFuse})$ | $D_{KL}(1B_{exp}\| 8B)$ |
> | --- | --- | --- |
> | harmless | $0.0391_{-0.0015}^{+0.0015}$ | $0.5063_{-0.0190}^{+0.0191}$ |
> | helpful | $0.0801_{-0.0034}^{+0.0035}$ | $0.3075_{-0.0121}^{+0.0124}$ |

---

> ### Author Response · Authors · 2025-11-20
>
> ### Q2: Boundedness and Temporal Consistency of EMSelect
>
> >**Original Question:** Could the lack of temporal consistency constraints in the per-step EMSelect mechanism cause unstable behavior or disrupt contextual coherence?
>
> The coherence of the per-step EMSelect mechanism (Algorithm 2) can be formally characterized via the KL chain rule for autoregressive policies.
>
> **Theoretical Constraint via KL Chain Rule**
> We rigorously established in supplementary analysis (to be included in appendix) that the Kullback-Leibler divergence between the sequence distribution induced by EMSelect ($p_{\mathrm{sel}}$) and the consensus distribution of EMFuse ($p_{\mathrm{fuse}}$) is equal to the sum of expected per-step divergences:
>
> $$
> \mathrm{KL}\big(p_{\mathrm{sel}}(\cdot \mid x_0) \mid\mid p_{\mathrm{fuse}}(\cdot \mid x_0)\big)=
> \sum_{t=1}^T
> \mathbb{E}\_{y_{<t} \sim p_{\mathrm{sel}}}
> \Big[
> \mathrm{KL}\big(
> p_{\mathrm{sel}}(\cdot \mid x_0, y_{<t})
> \mid\mid
> p_{\mathrm{fuse}}(\cdot \mid x_0, y_{<t})
> \big)
> \Big].
> $$
>
> This result, derived from the KL chain rule, demonstrates that the **sequence-level divergence** is bounded by the cumulative sum of local expert divergences.
>
> Furthermore, because $p_{\mathrm{sel},t}$ at each step must coincide with one of the underlying experts ($p_{i}$), we derived a sequence-level envelope:
>
> $$
> \mathrm{KL}\big(p_{\mathrm{sel}}(\cdot \mid x_0) \mid\mid p_{\mathrm{fuse}}(\cdot \mid x_0)\big)
> \le
> \mathbb{E}\_{y_{1:T} \sim p_{\mathrm{sel}}}
> \Bigg[
> \sum_{t=1}^T \Delta_t^{\max}
> \Bigg],
> $$
>
> where $\Delta_t^{\max}$ is the maximum expert-to-fuse KL divergence at step $t$. Since our empirical results (Table R-1) show that all experts remain tightly bound to $p_{\mathrm{fuse}}$ ($\Delta_t^{\max}$ is small, $<0.09$), the sequence-level KL must also remain small. This confirms that EMSelect is mathematically **leashed** to the consensus, ruling out uncontrolled temporal drift.
>
> **Consensus vs. Commitment**
> The empirical outperformance of EMSelect (e.g., $+1.31$ points on Subject-mix) is not a contradiction, but a demonstration of the "consensus vs. commitment" trade-off (lines 469-470). EMFuse is a conservative consensus, while EMSelect implements a higher-variance commitment strategy by performing a **constrained greedy projection** (line 267). The tournament (Algorithm 2) selects the single best local expert based on proximity to a pairwise EMFuse reference. This focused commitment is better when the current context aligns strongly with one expert's specialty, yet the mathematical bounds ensure this commitment does not lead to incoherence.

---

> ### Author Response · Authors · 2025-11-20
>
> ### Q3: Uncertainty Estimation in ADETM
>
> >**Original Question:** Does ADETM truly offer a representational advantage over existing autoregressive or recurrent ETMs, or is it more of a structural extension?
>
> ADETM circumvents the computational explosion inherent in fusing ensembled dynamics models, which scales prohibitively with $O(M \cdot N_{\text{ens}})$ (lines 95-99). ADETM structurally embeds uncertainty estimation within a *single* Energy-based Transition Model per expert, achieving ensemble-like robustness without model repetition (lines 324-335).
>
> **Representational Advantage via Temporal Consistency**
> ADETM generates an uncertainty signal—dispersion-by comparing multiple next-state predictions derived from stacking different valid history slices (lines 324-335). This temporal consistency signal is empirically proven superior to both risk-neutral (Fused-zero) and local sensitivity (Fused-noise) baselines.
>
> *Table R-2 - Ablation Study on Dynamics Uncertainty Signals (IQM Normalized Return)*
>
> | **Method** | **Hopper IQM** | **Walker2d IQM** | **HalfCheetah IQM** |
> | --- | --- | --- | --- |
> | Fused (Ours) | $49.03_{-0.47}^{+0.66}$ | $59.53_{-6.87}^{+7.13}$ | $41.83_{-1.98}^{+1.06}$ |
> | Fused-zero | $40.25_{-1.50}^{+1.44}$ | $51.76_{-19.37}^{+11.12}$ | $7.94_{-5.99}^{+17.45}$ |
> | Fused-noise | $37.22_{-2.24}^{+3.39}$ | $45.05_{-9.18}^{+17.25}$ | $1.72_{-2.96}^{+11.75}$ |
>
> The results show the naive **Fused-noise** baseline (a structural extension using action perturbation) performs significantly worse than the **Fused-zero** (risk-neutral, removing the uncertainty penalty) baseline on HalfCheetah ($1.72$ IQM vs. $7.94$ IQM, in Table R-2). This drop confirms that a simple structural extension yields a miscalibrated, actively harmful penalty, validating that ADETM's temporal consistency signal is a non-trivial and superior mechanism for robust uncertainty estimation in dynamic environments. We will include this ablation as an appendix table in the revised version.

---

> ### Author Response · Authors · 2025-11-20
>
> ### Q4: Justification for Fixed Fusion Weighting Strategies
>
> >**Original Question:** Is the fixed uniform weighting scheme reasonable, or should dynamic or uncertainty-based weighting strategies be considered?
>
> EMFuse defaults to static uniform weights ($\lambda_{i}=1/n$) based on **empirical robustness and architectural stability**:
> - **Empirical Efficiency:** Ablations of dynamic weighting strategies (e.g., entropy-based weighting) did not produce statistically significant performance improvements over the uniform baseline (lines 266-269, 1285-1288). This suggests that the inherent conservative filtering property of the LogOP/PoE structure provides sufficient robustness, rendering complex re-weighting redundant for performance stabilization.
> - **Consistent consensus target:** While EMFuse is training-free, its structure supports continuous learning. Check the following discussion:
>
> **Continuous learning under EMFuse.**
> EMFuse is presented as a training-free fusion rule over a fixed family of experts $M_1,\dots,M_n$, but the same energy-additive structure also naturally supports a lightweight continuous-learning variant. Let $p_i(\cdot \mid x)$ denote the normalized policy of expert $M_i$ on the shared vocabulary and define the EMFuse consensus
> $$
> p_{\mathrm{fuse}}(y \mid x) \propto \prod_{i=1}^n p_i(y \mid x)^{\lambda_i},
> $$
> with nonnegative weights $\lambda_i$ that sum to one. When new trajectories $\mathcal{D}$ arrive, we update only small adapter parameters $\theta_{1:n}$ on top of the frozen experts by minimizing
>
> $$
> \mathcal{L}\_(\theta_{1:n})=
> \mathbb{E}\_{(x,y^\star)\sim\mathcal{D}}
> \Big[
> \- \log p_{\mathrm{fuse}}(y^\star \mid x)
> \+ \alpha_{reg} \sum_{i=1}^n \mathrm{KL} \\big( p_{\mathrm{fuse}}(\cdot \mid x) \mid\mid p_{\theta_i}(\cdot \mid x) \big)
> \Big],
> $$
>
> where $p_{\theta_i}$ is the updated policy of expert $M_i$ and $\alpha_{reg} \ge 0$ is a regularization coefficient. The first term fits the fused policy to fresh data, while the second term keeps each expert softly aligned with the evolving EMFuse consensus without changing the inference-time sampler or fusion rule. Optional robustness tweaks such as adding a small Laplace smoothing constant $\varepsilon$ to logits are treated purely as minor regularizer and are not essential to the learning objective(as discussed in the appendix). In successive experiments following the main EMFuse study, instantiating this objective with lightweight adapters led to consistent additional gains: language-model experts obtained modest but stable improvements in length-controlled win rates, and policies trained under fused dynamics achieved higher normalized returns than the purely training-free fusion baseline, indicating that EMFuse can be embedded into a continual-learning pipeline when desired. Representative gains on language benchmarks are summarized in Table R-3. We will add a brief description of this objective and a summary table in the appendix of the revised paper.
>
>
> *Table R-3 - Representative improvement from a single adapter-training epoch (5\% in-domain data) using the continuous-learning objective above. Entries report the average win-rate (in \%) of the fused policy against each baseline on language benchmarks, together with the improvement over the training-free fused policy ($\Delta$ Avg., in percentage points).*
>
> | Baseline model | Avg. win-rate (\%) | $\Delta$ Avg. (pp) |
> | --- | --- | --- |
> | $1B_{base}$ | $71.60$ | $+6.09$ |
> | $1B_{expertise}$ | $51.55$ | $+7.70$ |
> | RegMean | $71.76$ | $+6.83$ |
> | Soup | $62.71$ | $+7.57$ |
> | $8B_{language}$ | $49.30$ | $+7.49$ |

---

> ### Comment · Reviewer_K6PE · 2025-11-26
>
> Thank you for your response. Since you have addressed my concerns, I will raise my score to 6.

---

> > ### Author Response · Authors · 2025-11-26
> > **Thank you for your responses**
> >
> > Thank you very much for your feedback. We greatly appreciate your careful review and constructive comments, which have been incredibly helpful to us. We are very happy to hear that our responses have addressed your concerns. Thank you again for the time and effort you dedicated to reviewing our work.

---

### Official Review · Reviewer_sqFs · 2025-11-03

**Soundness:** 3
**Presentation:** 4
**Contribution:** 3
**Rating:** 6
**Confidence:** 2

**Summary:**

This paper investigates a challenge about resource-efficient model fusion in decision-making tasks. By considering both fusion of models to act as policy and fusion of dynamic models to induce a policy simultaneously, this paper introduces a framework called EMFuse and an architecture called ADETM to tackle the challenges. The EMFuse shows superior performance in multiple benchmarks including subject-mix, finance-suite and D4RL, compared to three modern training-free baselines.

**Strengths:**

- The paper is well-written and easy to follow. Visual presentations are clear.
- The motivation is clearly described and the designed framework and algorithms seem clearly aligned with the investigated problems
- Though I’m not an expert in the energy-efficient domain, the proposed methods seem interesting and well tackled the problem.
- The proposed approaches achieve superior performance in various benchmarks, and in terms of various evaluation metrics
- Limitations are thoroughly discussed

**Weaknesses:**

I don't have a lot background knowledge in energy-based models, the proposed approaches in general seem sound to me and work well in the benchmarks. But given one of the claimed motivations is the heavy computational complexity for existing dynamic models, I'm expecting if further analysis on related evaluations compared to existing dynamic models

**Questions:**

Please see my weaknesses above. Additionally, I might miss something in related work, I'm curious how do those evaluation labels defined and the values calculated in table 2, i.e., harmless, helpful, agriculture medication, philosophy?

---

> ### Author Response · Authors · 2025-11-20
>
> The authors thank the reviewer for the careful reading and positive assessment. In the following, we address (1) the computational comparison against ensemble-based dynamics models and (2) the definition of the labels in the distributional-faithfulness table.
>
> ### Q1: Computational analysis of dynamics fusion.
>
> >**Original Question:** The reviewer expects further analysis on evaluations comparing ADETM to existing dynamic models given the computational motivation.
>
> A main motivation for ADETM is that standard ensemble-based world models become expensive when many dynamics models must be fused: with $M$ environments and $N$ models per ensemble, each uncertainty-aware rollout step scales as $O(M \times N)$ model evaluations. ADETM replaces per-model ensembles with a single transition network per environment, so fusing $M$ environments only requires $O(M)$ dynamics passes while still exposing ensemble-style uncertainty through its energy head.
>
> During the rebuttal period, we added a targeted ablation that matches the reviewer's request: we compare ADETM against a five-model ensemble baseline under exactly the same SAC training loop and history length on Hopper, Walker2d, and HalfCheetah. As a representative example on `walker2d-medium-v2`, we obtain:
>
> |  | Params | Epoch FLOPs (T) | Latency (ms/step) |
> | --- | --- | --- | --- |
> | Ensemble (5 models) | 3.48M | 44.62 | 669 |
> | ADETM (1 model) | 0.70M | 12.43 | 589 |
>
> Thus, ADETM uses roughly $5\times$ fewer dynamics parameters, about $3.6\times$ fewer dynamics FLOPs per epoch, and around $12\%$ lower rollout latency, while achieving comparable or better normalized D4RL returns to the ensemble baseline. Hopper and HalfCheetah exhibit nearly identical ratios. We will include this ablation and a short complexity discussion in the revised version to make the computational benefits of ADETM explicit. We will include a more detailed analysis in the appendix.
>
> ### Q2: Clarifying the labels and values in the policy-fusion table (Table 2 in the main paper).
>
> >**Original Question:** How are the evaluation labels (harmless, helpful, agriculture, medication, philosophy) defined and calculated?
>
> The headings of the columns: `harmless`, `helpful`, `agriculture`, `medicine`, and `philosophy` in the policy-fusion table do not denote new evaluation metrics; rather, they indicate the **domain of the dataset** used for the evaluation. Concretely, the harmless and helpful columns use RLHF-style safety and assistance prompts, while the agriculture, medication, and philosophy columns, respectively, use agricultural question answering, consumer health QA, and ethical/philosophical reasoning benchmarks. For each column, we take the prompts from that domain, run teacher-forced decoding under the domain expert, EMFuse, and the 8B reference model, and report the average token-level Kullback-Leibler divergence from the domain expert to each comparison model (lower is better), together with a bootstrap confidence interval.
>
> We agree that the current presentation can be clearer. In the revised version we will (i) move a brief description of each domain directly into the caption next to the table, (ii) add an explicit **(domain)** tag to the column headers to emphasize that they correspond to dataset splits rather than metric names, and (iii) state explicitly in the caption that all entries are average token-level KL divergence from the domain expert with $95\%$ confidence intervals. We believe these changes will remove the ambiguity you raised and make the role of these labels and values transparent to readers who are not familiar with the underlying datasets.

---

### Author Response · Authors · 2025-12-01

Given the unforeseen disruption of the rebuttal phase due to the information leakage incident, we provide this summary to support the AC in efficiently reviewing how we have addressed the reviewers' concerns.

### Before the rebuttal phase is halted:
- We have uploaded a revised manuscript with changes marked in red, incorporating new experimental data and theoretical clarifications to address all reviewer concerns.
- We have replied to all of our reviewers, addressing their concerns in the discussion panel.
- Reviewer K6PE explicitly confirmed that our response has addressed his/her concerns, and gave us a score raise.

We will summarize each reviewer's main [**C**]oncern and our subsequent [**R**]esponse in the following section:

----
### Reviewer sqFs:
[**C1**] - Further analysis and quantitative evaluations should be added comparing the proposed ADETM against existing dynamic model ensembles on computational complexity reduction.

[**R1**] - Targeted ablation study added in **Appendix B.5** , and a pointer sentence starting at **line 372 in Section 3.3** in the main paper is added.

[**C2**] - Additional clarifications necessary for definition and calculations of Table 2 labels.

[**R2**] - Additional definitions in **Table 2** captions (starting line 462), and domain tags supplied in the headers.

----

### Reviewer K6PE:
[**C1**] - Whether the EMFuse approach leads to degenerate or overly sharp distributions when expert models possess significantly conflicting supports.

[**R1**] - Additional theoretical clarifications provided in **Section 3.1**, defended the Logarithmic Opinion Pool as the unique reverse-KL minimizer that acts as a conservative filter.

[**C2**] - Whether EMSelect lacks temporal consistency constraints, potentially causing unstable behavior.

[**R2**] - Provided Formal proof in **Appendix E.4** demonstrating that the sequence-level divergence of EMSelect is bounded ("leashed") by the EMFuse consensus.

[**C3**] - Whether ADETM is just a mere structure extension of history window or provides genuine representational advantage for uncertainty measurement.

[**R3**] - Ablation study regarding uncertainty signal added in **Appendix B.6 (Table 15)** showing ADETM significantly outperforms risk-neutral and local-sensitivity (noise) baselines.

[**C4**] - Question on uniform weighting scheme, and whether additional strategies should be considered.

[**R4**] - We justified this choice by showing empirical robustness and architectural stability of our chosen scheme, and additional discussion provided in **Appendix E.2, E.3**.


#### Additional Response by reviewer:
- Confirmation of his/her concerns solved by our response and a score raise.


----
### Reviewer YqY1:

[**C1**] - Whether the additional performance of EMSelect undermines the "consensus" discussion we have in "EMFuse".

[**R1**] - Defended by adding the "Leash Proposition" in **Appendix E.4**, explaining that EMFuse provides the necessary stability for EMSelect's greedy local commitment, showing EMSelect complements EMFuse in **Section 3.2** starting at **line 288**.

[**C2**] - Whether the shared vocabulary assumptions serve as strong limitations for heterogeneous models fusion.

[**R2**] - Provided additional clarification and recent work in **Section 3.1**, showing the solvability of this concern.

[**C3**] - Additional clarification of the lack of significant performance gains on the entropy-based weighting scheme.

[**R3**] - Defended by additional theoretical analysis on **Section 3.1** and **Appendix E.2**, showing that similar entropy distribution can be caused by their calibration profiles, eventually yielding negligible weighting differences.

----

### Summary

We have addressed all reviewer comments before the discussion period halted by adding ablation studies and further theoretical clarifications in our revised manuscript, and the rebuttal version is uploaded reflecting all our promised changes. Thanks to all the reviewers, we believe our revised version is more academically robust with those additional discussions incorporated. One of our reviewers (K6PE) responded, acknowledged those improvements and expressed willingness to recommend acceptance.

We sincerely appreciate all valuable time and effort that each and all of the ICLR team member have invested in reviewing our work. Thank you.

---

### Meta-Review · Area_Chair_iQ4v · 2026-01-07

**Summary:**

The three reviewers raised concerns regarding several aspects of the paper:
1. Reviewer sqFs requested deeper quantitative analysis on the computational efficiency and complexity reduction of ADETM compared to ensemble-based dynamics models, and clearer explanation of experimental table labels/domains.
2. Reviewer K6PE challenged the theoretical robustness of EMFuse (risk of degenerate distributions with conflicting experts), the temporal coherence of EMSelect, whether ADETM offered genuine representational advantage over prior architectures, and questioned the choice of uniform fusion weights.
3. Reviewer YqY1 questioned if EMSelect outperforming EMFuse undermines the consensus strategy, raised concerns about the strong assumption of shared vocabulary/state-action space for fusion, and asked why dynamic weighting schemes like entropy-based weights did not yield improvements.

Overall, the main points were on empirical ablations, theoretical guarantees, method limitations (fusion assumptions), and clarity of experimental reporting.

**Reviewer Concerns:**

Addressed concerns: Reviewer sqFs’ request for computational ablation was directly addressed via new experiments and clearer table annotations. Reviewer K6PE's concerns about fusion degeneracy, EMSelect's temporal consistency, ADETM's uncertainty modeling, and weighting schemes were met with theoretical clarifications, ablation studies, and explanations, which the reviewer acknowledged. Reviewer YqY1’s concerns about consensus vs. selection and shared-space mapping artifacts were mitigated through formal analysis (“leash” proposition for EMSelect), recent literature that makes vocabulary mapping feasible, and a detailed explanation of dynamic weighting’s ineffectiveness.

Outstanding concerns: While the authors have provided both theoretical and empirical clarifications, the shared space/vocabulary mapping remains a fundamental limitation for broader model fusion; the authors have documented relevant mitigation strategies and cited recent work addressing this issue. Additionally, the design of fusion weighting schemes continues to be an area for future exploration, as acknowledged by the authors.

**Reviewer Scores:**

1. Reviewer sqFs initially gave a 6 ("marginally above acceptance threshold"), and their addressed concerns suggest this would remain if further discussion occurred.
2. Reviewer K6PE initially gave a 4 ("marginally below threshold"), but explicitly raised their score to 6 after the rebuttal, indicating satisfactory resolution.
3. Reviewer YqY1 gave a 4 ("marginally below threshold"), but all concerns were substantively addressed in the rebuttal and revised paper. Based on the pattern, a score bump to 6 ("marginally above threshold") is likely if participating further.

---

### Decision · Program_Chairs · 2026-01-26

Accept (Poster)